# Class-agnostic Reconstruction of Dynamic Objects from Videos

**Zhongzheng Ren**,*   **Xiaoming Zhao**,*   **Alexander G. Schwing**
University of Illinois at Urbana-Champaign
https://jason718.github.io/redo

## Abstract

We introduce **REDO**, a class-agnostic framework to **RE**construct the **D**ynamic **O**bjects from RGBD or calibrated videos. Compared to prior work, our problem setting is more realistic yet more challenging for three reasons: 1) due to occlusion or camera settings an object of interest may never be entirely visible, but we aim to reconstruct the complete shape; 2) we aim to handle different object dynamics including rigid motion, non-rigid motion, and articulation; 3) we aim to reconstruct different categories of objects with one unified framework. To address these challenges, we develop two novel modules. First, we introduce a canonical 4D implicit function which is pixel-aligned with aggregated temporal visual cues. Second, we develop a 4D transformation module which captures object dynamics to support temporal propagation and aggregation. We study the efficacy of REDO in extensive experiments on synthetic RGBD video datasets SAIL-VOS 3D and DeformingThings4D++, and on real-world video data 3DPW. We find REDO outperforms state-of-the-art dynamic reconstruction methods by a margin. In ablation studies we validate each developed component.

## 1   Introduction

4D (3D space + time) reconstruction of both the geometry and dynamics of different objects is a long-standing research problem, and is crucial for numerous applications across domains from robotics to augmented/virtual reality (AR/VR). However, complete and accurate 4D reconstruction from videos remains a great challenge for mainly three reasons: 1) partial visibility of objects due to occlusion or camera settings (*e.g.*, out-of-view parts, non-observable surfaces); 2) complexity of the dynamics including rigid-motion (*e.g.*, translation and rotation), non-rigid motion (deformation caused by external forces), and articulation; and 3) variability within and across object categories.

Existing work addresses the above challenges by assuming complete visibility through a multi-view setting [33, 4], or by recovering only the observable surface rather than the complete shape of an object [57], or by ignoring rigid object motion and recovering only the articulation [59], or by building shape templates or priors specific to a particular object category like humans [47]. However, these assumptions also limit applicability of models to unconstrained videos in the wild, where these challenges are either infeasible or only met when taking special care during a video capture.

In contrast, we aim to study the more challenging unconstrained 4D reconstruction setting where objects may never be entirely visible. Specifically, we deal with visual inputs that suffer from: 1) *occlusion:* the moving occluder and self-articulation cause occlusion to change across time; 2) *cropped view:* the camera view is limited and often fails to capture the complete and consistent appearance across time; 3) *front-view only:* due to limited camera motion, the back side of the objects are often not captured at all in the entire video. Moreover, we focus on different dynamic object-types

---

*Indicates equal contribution

35th Conference on Neural Information Processing Systems (NeurIPS 2021).

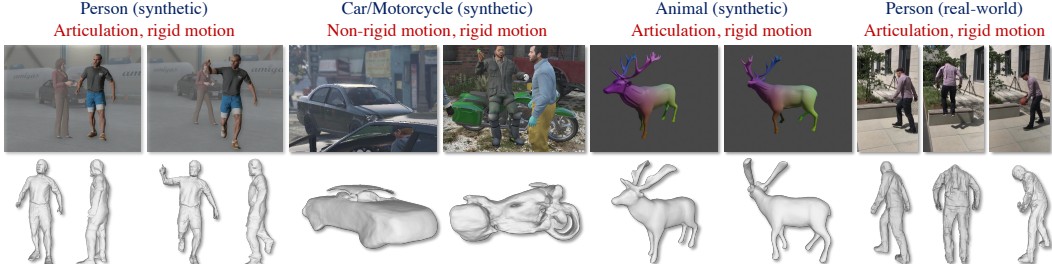

| Person (synthetic) | Car/Motorcycle (synthetic) | Animal (synthetic) | Person (real-world) |
| Articulation, rigid motion | Non-rigid motion, rigid motion | Articulation, rigid motion | Articulation, rigid motion |

Figure 1: We present REDO, a 4D reconstruction framework that predicts the geometry and dynamics of various objects for a given video clip. Despite objects being either occluded (*e.g.*, car/motorcycle) or partially-observed, REDO recovers relatively complete and temporally smooth results.

with complex motion patterns. These objects could either move in 3D space, or be deformed due to external forces, or articulate themselves. Importantly, we aim for a class-agnostic reconstruction framework which can recover the accurate shape at each time-step.

To achieve this we develop REDO. As illustrated in Fig. 1, REDO predicts the shape of different objects (*e.g.*, human, animals, car) and models their dynamics (*e.g.*, articulation, non-rigid motion, rigid motion) given input video clips. Besides the RGB frames, REDO takes as input the depth map, masks of the objects of interest, and camera matrices. In practice these inputs are realistic as depth-sensors are increasingly prevalent [19, 78] and as segmentation models [44, 17] are increasingly accurate and readily available, *e.g.*, on mobile devices. If this data isn't accessible, off-the-shelf tools are applicable (*e.g.*, SfM [71], instance segmentation [26], video depth estimation [51]).

To address the partial visibility challenge introduced by occlusion or camera settings, REDO predicts a temporally coherent appearance in a canonical space. To ensure that the same model is able to reconstruct very different object types in a unified manner we introduce a pixel-aligned 4D implicit representation, which encourages the predicted shapes to closely align with the 2D visual inputs (§ 3.1). The visible parts from different frames of the video clip are aggregated to reconstruct the same object (§ 3.2). During inference, the reconstructed object in canonical space is propagated to other frames to ensure a temporally coherent prediction (§ 3.3).

REDO achieves state-of-the-art results on various benchmarks (§ 4). We first conduct experiments on two synthetic RGBD video datasets: SAIL-VOS 3D [27] and DeformingThings4D++ [42]. REDO improves over prior 4D reconstruction work [59, 70] by a great margin (+5.9 mIoU, -0.085 mCham., -0.22 mACD on SAIL-VOS 3D and +2.2 mIoU, -0.063 mCham., -0.047 mACD on DeformingThings4D++ over OFlow [59]). We then test on the real-world calibrated video dataset 3DPW [86]. We find that REDO generalizes well and consistently outperforms prior 4D reconstruction methods (+10.1 mIoU, -0.124 mCham., -0.061 mACD over OFlow). We provide a comprehensive analysis to validate the effectiveness of each of the introduced components.

## 2   Related work

In this section, we first discuss possible geometry representations. We then review fusion-based and learning-based 4D reconstruction approaches, followed by a brief introduction of Motion Capture methods. Lastly, we discuss works of dynamics modelling and related 4D reconstruction datasets.

**Geometric representations.** Representations to describe 3D objects can be categorized into two groups: discrete and continuous. Common discrete representations are voxel grids [22, 90, 8], octrees [68, 82], volumetric signed distance functions (SDFs) [10, 30, 56], point-clouds [1, 18, 67], and meshes [23, 34, 55, 38, 88]. Even though being widely used, these representations pose important challenges. Voxel grids and volumetric-SDFs can be easily processed with deep learning frameworks, but are memory inefficient [76, 52, 64]. Point-clouds are more memory efficient to process [65, 66], but do not contain any surface information and thus fail to capture fine details. Meshes are more expressive, but their topology and discretization introduce additional challenges.

To overcome these issues, continuous representations, *i.e.*, parametric implicit functions, are introduced to describe the 3D geometry of objects [7, 60, 53] and scenes [54, 73]. These methods are not constrained by discretization and can thus model arbitrary geometry at high resolution [70, 81]. In

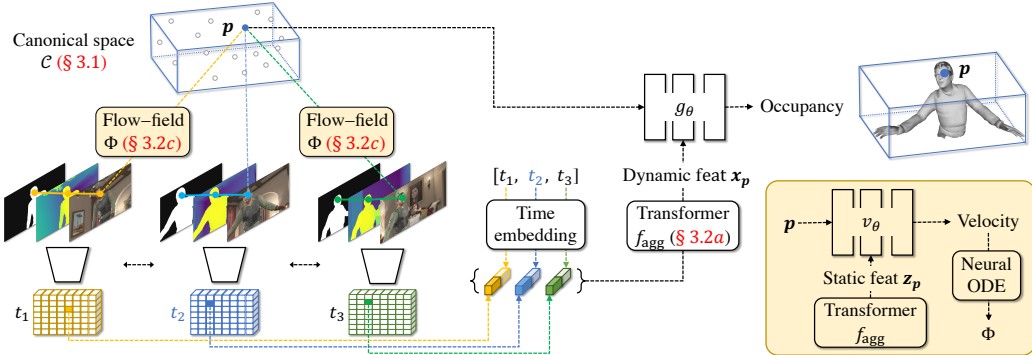

Figure 2: **Framework overview.** For a query point **p** in canonical space, REDO first computes the pixel-aligned features from the feature maps of different frames using the flow-field $\Phi$. It then aggregates these features using the temporal aggregator $f_{\text{agg}}$. The obtained dynamic feature $\mathbf{x_p}$ is eventually used to compute the occupancy score for shape reconstruction.

practice, discretized representations like a mesh can be easily extracted from implicit functions via algorithms like Marching Cubes [49].

**Fusion-based 4D reconstruction.** DynamicFusion [57] is the pioneering work for reconstructing non-rigid scenes from depth videos in real-time. It fuses multiple depth frames into a canonical frame to recover the observable surface and adopt a dense volumetric 6D motion field for modeling motion. Another early work [14] reconstructs dynamic objects without explicit motion fields. Improvements of DynamicFusion leverage more RGB information [28], introduce extra regularization [74, 75], develop efficient representations [20], and predict additional surface properties [24]. Importantly, different from the proposed direction, these works only recover the observable surfaces rather then the complete shape. Moreover, these works often fail to handle fast motion and changing occlusion. Note, these models often have no trainable parameters and are more geometry-based.

**Learning-based 4D reconstruction.** Reconstruction of the complete shape of dynamic objects is considered hard due to big intra-class variations. For popular object categories such as humans or certain animals, supervised learning of object template parameters has been utilized to ease reconstruction [36, 94, 47, 96, 97]. These templates are carefully designed parametric 3D models, which restrict learning to a low-dimensional solution space. However, the expressiveness of templates is limited. For instance, the SMPL [47] human template struggles to capture the clothing or hair styles. In addition, systems relying on templates are class-specific as different object categories need to be parameterized differently. We think it remains illusive to construct a template for every object category. To overcome these issues, OFlow [59] directly learns 4D reconstruction from scratch. However, it predicts articulation in normalized model space and thus overlooks rigid-motion of the object. It also struggles to handle occlusion. When ground-truth 3D models are not available, dynamic neural radiance fields (NeRF) [54, 43, 63, 61] learn an implicit scene representation from videos, but are often scene-specific and don't scale to a class-agnostic setting. Self-supervised methods [91, 35, 41] are promising and learn 4D reconstruction via 2D supervision of differentiable rendering [37, 48]. In contrast, in this work, we present a class-agnostic and template-free framework which learns to recover the shape and dynamics from input videos.

**Motion/Performance capture.** When restricting ourselves to human modeling, 4D reconstruction is often referred to as motion capture (MoCap). Marker and scanner based MoCap systems [62, 46, 40] are popular in industry, yet inconvenient and expensive. Without expensive sensors, camera-based methods are more applicable but often less competitive. Standard methods rely on a multi-view camera setup and leverage 3D human body models [32, 95, 21, 85]. Active research problems in the MoCap field include template-free methods [16, 15, 11], single-view depth camera methods [2, 92, 93], monocular RGB video methods [25, 79], and fine-grained body parts (*e.g.*, hand, hair, face) methods [72, 50, 89, 33]. In contrast, this paper studies a more generic class-agnostic setting where multiple different non-rigid objects are reconstructed using one unified framework.

**Dynamics modeling.** In different communities, the task of dynamics modeling is named differently. For 2D dynamics in images, optical flow methods [13, 80, 83] have been widely studied and used. Scene flow methods [43, 45, 31] aim to capture the dynamics of all points in 3D space densely, and are often time-consuming and inefficient. For objects, non-rigid tracking or matching methods [4, 77, 9, 5]

study the dynamics of non-rigid objects. However, these methods often only track the observable surface rather then the complete shape space. Recently, neural implicit functions [43, 59] have been applied to estimate 3D dynamics, which we adapt and further improve through conditioning on pixel-aligned temporally-aggregated visual features.

**4D reconstruction datasets.** To support 4D reconstruction, a dataset needs to have both ground-truth 3D models and accurate temporal correspondences. Collecting such a dataset in a real-world setting is extremely challenging as it requires either expensive sensors or restricted experimental settings. To simplify the setting, existing data is either class-specific (*e.g.*, human) [29, 87, 3], or of extremely small scale [74, 28], or lacking ground-truth 3D models [5]. Given the progress in computer graphics, synthetic data is becoming increasingly photo-realistic and readily available. DeformingThings4D [42] provides a large collections of humanoid and animal 4D models, but lacks textures and background. SAIL-VOS 3D [27] contains photo-realistic game videos together with various ground-truth. For more details, please see the comparison table in Appendix § B.

## 3 REDO

We aim to recover the 3D geometry of a dynamic object over space and time given a RGBD video together with instance masks and camera matrices. To achieve this we develop REDO which is illustrated in Fig. 2. Specifically, REDO reconstructs an object shape and its dynamics in a canonical space, which we detail in § 3.1. For this REDO employs a temporal feature aggregator, a pixel-aligned visual feature extractor, and a flow-field, all of which we discuss in § 3.2. These three modules help align dynamics of the object across time and resolve occlusions to condense the most useful information into the canonical space. Lastly, we detail inference (§ 3.3) and training (§ 3.4).

**Notation.** The input of REDO is a fixed-length video clip denoted via $\{I_1, \ldots, I_N\}$. It consists of $N$ RGBD frames $I_i \in [0, 1]^{4 \times W \times H}$ ($i \in \{1, \ldots, N\}$) of width $W$ and height $H$, each recorded at time-step $t_i$. For each frame $I_i$, we also assume the camera matrix and instance mask $m_{ij} \in \{0, 1\}^{W \times H}$ are given, where $m_{ij}$ indicates the set of pixels that correspond to object $j$. For readability, we define the following operations: 1) `propagate`: transforms temporally in 3D; 2) `project`: transforms from 3D to 2D (image space); and 3) `lift`: transforms from 2D (image space) to 3D.

### 3.1 Canonical 4D implicit function

Dynamic objects deform and move across time and are often occluded or partially out-of-view. In addition, the depth-size ambiguity also challenges reconstruction algorithms. To condense information about the object across both space and time, we leverage a canonical space $\mathcal{C} \subseteq \mathbb{R}^3$, which aims to capture a holistic geometry representation centered around the object of interest. In our case $\mathcal{C}$ denotes a volume-constrained 3-dimensional space. Temporally, the canonical space corresponds to the center frame $I_c$ at time-step $t_c$ with $c = \lceil (1 + N)/2 \rceil$.

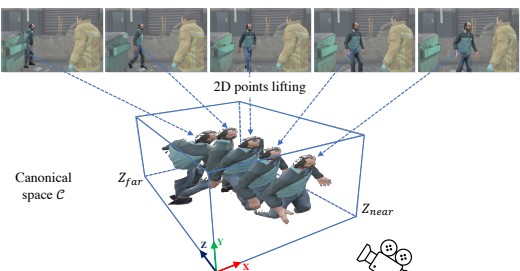

Figure 3: **Canonical Space.** Depth points from different frames are lifted and aggregated to form an enlarged space for capturing the dynamic objects.

To construct the canonical space, we infer a 3D volume around the dynamic object $j$. For all frames $I_i$, $i \in \{1, \ldots, N\}$, we first `lift` the pixels of instance mask $m_{ij}$ of object $j$ to the world space using the depth map and the camera matrices. These 3D points are then aggregated into canonical space using the camera matrix of the canonical frame. From the aggregated point clouds we infer the horizontal and vertical bounding box coordinates as well as the object's closest distance from the camera, *i.e.*, $Z_{\text{near}}$. However, since the visible pixels only represent the front surface of the observed object, the largest distance of the object from the camera, *i.e.*, $Z_{\text{far}}$, is unknown. To estimate this value, we simply set $Z_{\text{far}}$ to stay a fixed distance from $Z_{\text{near}}$. We illustrate an example of this canonical space in Fig. 3, where the inferred canonical space (blue volume) is relatively tight and big enough to capture the complete shape and dynamics of the observed object.

To more precisely capture an object inside the volume-constrained canonical space, we use a 4D implicit function. For each 3D query point in canonical space, this function conditions on a temporal feature to indicate whether that point is inside or outside the object. Specifically, for a query point

inside the canonical space, *i.e.*, for $\mathbf{p} \in \mathcal{C}$, the 4D implicit function is defined as:

$$g_\theta(\mathbf{p}, \mathbf{x_p}) : \mathcal{C} \times \mathbb{R}^K \to [0, 1]. \tag{1}$$

Intuitively, the higher the score $g_\theta(\cdot, \cdot)$, the more likely the point $\mathbf{p}$ is part of the object. Here, $\theta$ subsumes all trainable parameters in the framework and $\mathbf{x_p} \in \mathbb{R}^K$ is a $K$-dimensional dynamic feature which summarizes information from all input frames. Concretely,

$$\mathbf{x_p} = f_{\mathrm{agg}} \left( \{ f_{\mathrm{enc}}(\psi \left( \Phi(\mathbf{p}, t_c, t_i, v_\theta) \right), I_i) \}_{i=1}^N \right), \tag{2}$$

where $\psi(\cdot) : \mathcal{C} \to \mathbb{R}^3$ transforms a point in canonical space to world space. Meanwhile, $\Phi$ is the flow-field which `propagates` the point $\mathbf{p}$ from $t_c$ to $t_i$ in canonical space, leveraging a learned 3-dimensional velocity-field $v_\theta$. Note, $t_c$ refers to the time-step of the canonical frame $I_c$ while $t_i$ denotes the time-step of frame $I_i$. Given the transformed point $\psi(\Phi(\mathbf{p}, t_c, t_i, v_\theta)) \in \mathbb{R}^3$, we use a pixel-aligned feature encoder $f_{\mathrm{enc}}$ to extract a set of visual representations, each of which comes from one of the $N$ frames $I_i$, *i.e.*, $\forall i \in \{1, \ldots, N\}$. The set function $f_{\mathrm{agg}}$ is a temporal feature aggregator which merges the temporal information of different time-steps. We'll discuss details of temporal aggregator $f_{\mathrm{agg}}$, encoder $f_{\mathrm{enc}}$, flow-field $\Phi$ and velocity-field $v_\theta$ in § 3.2.

This canonical 4D implicit function `propagates` points across time and extracts the pixel-aligned visual representations from different frames. This differs from: 1) prior works that only consider static objects. Actually, setting $N = 1$ in Eq. (2) will simplify Eq. (1) to $g(\mathbf{p}, f_{\mathrm{enc}}(\psi(\mathbf{p}), I))$, recovering a static Pixel-aligned Implicit Function (PIFu) used in [69, 70]; 2) OFlow [59] which encodes the whole video clip $\{I_1, \ldots, I_N\}$ into one single feature vector, thus loosing spatial information.

## 3.2 Framework design

In this section, we introduce the temporal aggregator $f_{\mathrm{agg}}$, the feature extractor $f_{\mathrm{enc}}$, and the flow-field $\Phi$ used in Eq. (2). These modules help REDO aggregate information from different time-steps (frames) and model the complex dynamics.

**a) Temporal aggregator $f_{\mathrm{agg}}$.** To deal with partial visibility caused by occlusions or camera settings, we develop a transformer-based temporal aggregator $f_{\mathrm{agg}}$ as shown in Fig. 2. $f_{\mathrm{agg}}$ is a set function which computes $K$-dimensional point features $\mathbf{x_p} \in \mathbb{R}^K$. Assume the flow-field $\Phi$ is given and object $j$ is of interest. We first `propagate` a query point within the canonical space, *i.e.*, the point $\mathbf{p} \in \mathcal{C}$, to every time-step $t_i$, obtaining locations $\psi(\Phi(\mathbf{p}, t_c, t_i, v_\theta)) \in \mathbb{R}^3 \ \forall i \in \{1, \ldots, N\}$. We then `project` each 3D point $\psi(\Phi(\mathbf{p}, t_c, t_i, v_\theta))$ back to the corresponding 2D image frame $I_i$ using the associated camera matrix. For all points that are projected into the mask area $m_{ij}$, we extract a visual representation using the pixel-aligned feature extractor $f_{\mathrm{enc}}$, which we detail below. Note that due to partial visibility, we may not be able to extract features from every frame in the clip. To cope with this, the aggregator $f_{\mathrm{agg}}$ is designed as a transformer-based [84] set function. For more information, please see the implementation details in § 4.1 and Appendix § A.

**b) Pixel-aligned feature extractor $f_{\mathrm{enc}}$.** Pixel-aligned features help REDO make 3D predictions that are aligned with the visual 2D input. To achieve this, we develop $f_{\mathrm{enc}}(\mathbf{q}, I_i)$ where the first argument is a point $\mathbf{q}$ in 3D world space and the second argument is a frame $I_i$. We first `project` the point $\mathbf{q}$ in world space to the frame $I_i$, and then use a pre-trained convolutional neural net [58] to extract the 2D feature map of a video frame $I_i$. For points that fall within the instance mask of a frame, we extract its visual representation using bi-linear interpolation at the `projected` location. We then append a positional encoding [84] of the frame time-step $t_i$ to this feature which helps to retain temporal information. We illustrate this process in Fig. 2. The resulting feature is combined with visual cues from other frames to serve as input to the temporal aggregator $f_{\mathrm{agg}}$.

**c) Flow-field $\Phi$.** The flow-field models object dynamics in space and time. For this let $\Phi(\mathbf{p}, t_1, t_2, v_\theta) \in \mathcal{C}$ denote a flow-field function in canonical space. It computes the position at time-step $t_2$ of a 3D point, whose location is $\mathbf{p} \in \mathcal{C}$ at time-step $t_1$. To compute the displacement, we define a velocity field $v_\theta(\cdot)$ which represents the 3D velocity vectors in space and time via

$$v_\theta(\mathbf{p}, \mathbf{z_p}, t) : \mathcal{C} \times \mathbb{R}^K \times \mathbb{R} \to \mathcal{C}. \tag{3}$$

Here, $\mathbf{p} \in \mathcal{C}$ is a point in canonical space with corresponding static feature $\mathbf{z_p} \in \mathbb{R}^K$ computed as

$$\mathbf{z_p} = f_{\mathrm{agg}} \left( \{ f_{\mathrm{enc}}(\psi(\mathbf{p}), I_i) \}_{i=1}^N \right). \tag{4}$$

Here, $f_{\text{enc}}(\psi(\mathbf{p}), I_i)$ is the feature of world coordinate point $\psi(\mathbf{p})$ extracted from frame $I_i$. Note, $\mathbf{z_p}$ differs from $\mathbf{x_p}$ defined in Eq. (2). The feature $\mathbf{z_p}$ summarizes information of *static* locations from all frames $I_i$. This feature is beneficial as it helps capture whether a point remains static or whether it moves. The velocity network then leverages this feature $\mathbf{z_p}$ to predict object dynamics.

Using the velocity field, we compute the target location at time-step $t_2$ of a point originating from location $\mathbf{p}$ at time-step $t_1$ by integrating the velocity field over the interval $[t_1, t_2]$ via

$$\Phi(\mathbf{p}, t_1, t_2, v_\theta) = \mathbf{p} + \int_{t_1}^{t_2} v_\theta(\Phi(\mathbf{p}, t_1, t, v_\theta), \mathbf{z_p}, t)dt. \tag{5}$$

Note that $\Phi(\mathbf{p}, t_1, t_2, v_\theta)$ can represent both forward ($t_2 > t_1$) or backward motion ($t_2 < t_1$) given the initial location and velocity-field. To solve the flow-field for discrete video time-steps, we approximate the above continuous integral equation using a neural-ODE solver [6].

### 3.3   Inference

To reconstruct objects densely in a clip, the reconstruction/inference procedure is summarized as:

**Step 1:** We construct the canonical space at the center frame $I_c$ and sample a query point set $\mathcal{P} \subseteq \mathcal{C}$ uniformly from the inferred space. This step requires no network inference and is very efficient.

**Step 2:** We extract static features $\mathbf{z_p}$ for all $\mathbf{p} \in \mathcal{P}$ using Eq. (4). Next, we compute the trajectory of point $\mathbf{p}$, *i.e.*, $\Phi(\mathbf{p}, t_c, t_i, v_\theta)$ for all time-steps $t_i$ associated with frames $I_i$, $\forall i \in \{1, \ldots, N\}$, using a neural ODE solver which solves Eq. (5).

**Step 3:** We then compute dynamic features $\mathbf{x_p}$ for all $\mathbf{p} \in \mathcal{P}$ using Eq. (2).

**Step 4**: Using Eq. (1) we finally compute the occupancy scores for all $\mathbf{p} \in \mathcal{P}$, which are then transformed into a triangle mesh via Multi-resolution Iso-Surface Extraction (MISE) [53]. Note, the mesh is constructed in canonical space.

**Step 5**: To obtain the mesh associated with frame $I_i$, following Step 2, we use the flow-field $\Phi$ to `propagate` all vertices of the extracted mesh from the time-step $t_c$ of the canonical space to time-step $t_i$ which corresponds to non-canonical frame $I_i$. After using the function $\psi(\cdot)$ defined in Eq. (2), we obtain the mesh in 3D world space. While various discrete representations could be obtained from our implicit representation, we use meshes for evaluation and visualization purposes.

### 3.4   Training

REDO is fully differentiable containing the following parametric components: the temporal aggregator $f_{\text{agg}}$, the feature extractor $f_{\text{enc}}$, the velocity-field network $v_\theta$, and the reconstruction network $g_\theta$. For simplicity, we use $\theta$ to subsume all trainable parameters of REDO. To better extract shape and dynamics from given video clips, we train REDO end-to-end using

$$\min_\theta \mathcal{L}_{\text{shape}}(\mathcal{D}, \theta) + \mathcal{L}_{\text{temp}}(\mathcal{D}, \theta), \tag{6}$$

where $\mathcal{D}$ is the training set. $\mathcal{L}_{\text{shape}}(\mathcal{D}, \theta)$ is the shape reconstruction loss in canonical space which encourages REDO to recover the accurate 3D geometry. $\mathcal{L}_{\text{temp}}(\mathcal{D}, \theta)$ is a temporal coherence loss defined on temporal point correspondences. This loss encourages the flow-field $\Phi$ to capture the precise dynamics of objects. We detail training set, sampling procedure, and both losses next.

**Training set.** We train REDO on a dataset $\mathcal{D}$ that consists of entries from different videos and of various object instances. Formally, $\mathcal{D} = \left\{ \left( \{(I_i, t_i)\}_{i=1}^N, \mathcal{V}, \mathcal{Y} \right) \right\}$. Specifically, a data entry for an $N$-frame video clip includes three components: 1) a set of RGBD frames $\{I_i\}$, associated time-steps $\{t_i\}$, as well as corresponding camera matrices $\forall i \in \{1, \ldots, N\}$; 2) an instance ground-truth mesh $\mathcal{V}$ that provides-temporally aligned supervision. For every vertex $\mathbf{v} \in \mathcal{V}$, we use $\mathbf{v}_i$ to denote its position at time-step $t_i$ ; 3) the ground-truth occupancy $\mathcal{Y}$ in the canonical space. Concretely, for a point $\mathbf{p} \in \mathcal{C}$ in canonical space, occupancy label $y(\mathbf{p}) \in \{0, 1\}$ indicates whether $\mathbf{p}$ is inside the object ($y = 1$) or outside the object ($y = 0$). As mentioned in § 3.1, the canonical space is chosen so that it corresponds to time-step $t_c$, where $c = \lceil (1 + N)/2 \rceil$.

**Sampling procedure.** To optimize for the parameters $\theta$ during training we randomly sample a set of points $\mathbf{p} \in \mathcal{P}(\mathcal{V})$ within the canonical space. $\mathcal{P}(\mathcal{V})$ contains a mixture of uniform sampling and

importance sampling around the ground-truth mesh's surface $\mathcal{V}$ at time-step $t_c$. Similar strategies are also used in prior works [69, 70].

**Shape reconstruction loss.** To encourage that the canonical 4D implicit function $g_\theta$ accurately captures the shape of objects we use the shape reconstruction loss

$$\mathcal{L}_{\text{shape}}(\mathcal{D}, \theta) = \sum_{\left(\{(I_i, t_i)\}_{i=1}^N, \mathcal{V}, \mathcal{Y}\right) \in \mathcal{D}} \frac{1}{|\mathcal{P}(\mathcal{V})|} \sum_{\mathbf{p} \in \mathcal{P}(\mathcal{V})} \text{BCE}\left(g_\theta(\mathbf{p}, \mathbf{x_p}), y(\mathbf{p})\right), \tag{7}$$

where $\text{BCE}(\cdot, \cdot)$ represents the standard binary cross-entropy loss.

**Temporal coherence loss.** REDO models dynamics of objects explicitly through the flow-field $\Phi$, which leverages the velocity-field network $v_\theta$. As the ground-truth correspondences across time are available in $\mathcal{V}$, we define the temporal correspondence loss via the squared error

$$\mathcal{L}_{\text{temp}}(\mathcal{D}, \theta) = \sum_{\left(\{(I_i, t_i)\}_{i=1}^N, \mathcal{V}, \mathcal{Y}\right) \in \mathcal{D}} \frac{1}{N|\mathcal{V}|} \sum_{\mathbf{v} \in \mathcal{V}} \sum_{i=1}^N \|\Phi(\mathbf{v}_c, t_c, t_i, v_\theta) - \mathbf{v}_i\|_2^2. \tag{8}$$

## 4 Experiments

We first introduce the key implementation details (§ 4.1) and the experimental setup (§ 4.2), followed by the quantitative results (§ 4.3), qualitative results (§ 4.5), and an in-depth analysis (§ 4.4).

### 4.1 Implementation details

We briefly introduce the key implementation details. Check Appendix § A for a more detailed version.

**Input:** We assume all input clips are trimmed to have $N = 17$ frames following [59]. During training, clips are randomly sampled from original videos which have any length. For videos that are shorter than 17 frames, we pad at both ends with duplicated starting and ending frames to form the clips. The validation and test set consist of fixed 17-frame clips. This simplified input setting allows us to split development into manageable pieces, and allows fair comparison with prior work. In practice, dense reconstruction on the entire video is achieved via a sliding window method.

**Reconstruction network:** Following [69], the reconstruction network $g_\theta$ is implemented as a 6-layer MLP with dimensions (259, 1024, 512, 256, 128, 1) and skip-connections. The first layer's dimension of 259 is due to the concatenation of visual features (256-dim) and query point locations (3-dim).

**Temporal aggregator:** $f_{\text{agg}}$ uses a transformer model with 3 multi-headed self-attention blocks and a 1-layer MLP. Group normalization and skip-connections are applied for each block, and we set the hidden dimension to be 128. To compute the time-encoding, we use positional-encoding [84] with 6 exponentially increasing frequencies.

**Feature extractor:** $f_{\text{enc}}$ is implemented as a 2-stack hourglass network [58] following PIFu [69]. Given the instance mask, we take out the object of interest in the picture and resize it to $256 \times 256$ before providing it as input to $f_{\text{enc}}$. The output feature map has dimensions of $128 \times 128 \times 256$ of spatial resolution $128 \times 128$ and feature dimension $K = 256$.

**Velocity-field network:** $v_\theta$ uses a 4-layer MLP with skip-connections following [59], where the internal dimension is fixed to 128. It takes query points as input and adds the visual features to the activations after the 1st block. For ODE solvers, we use the Dormand–Prince method (dopri5) [12].

**Training:** In each training iteration we sample 2048 query points for shape reconstruction and 512 vertices for learning of temporal coherence. We train REDO end-to-end using the Adam optimizer [39] for 60 epochs with a batch size of 8. The learning rate is initialized to 0.0001 and decayed by $10\times$ at the 40th and 55th epochs.

### 4.2 Experimental setup

**Dataset.** We briefly introduce the three datasets used in our experiments below. For more details (*e.g.*, preparation, statistics, examples), please check Appendix § B.

| Dataset | SAIL-VOS 3D | | | DeformingThings4D++ | | | 3DPW | | |
|---------|-------------|--------|--------|--------|--------|--------|--------|--------|--------|
| Metrics | mIoU↑ | mCham.↓ | mACD↓ | mIoU↑ | mCham.↓ | mACD↓ | mIoU↑ | mCham.↓ | mACD↓ |
| *Static reconstruction* | | | | | | | | | |
| ONet [53] | 24.5 | 0.951 | - | **60.2** | **0.260** | - | 29.8 | 0.440 | - |
| PIFuHD [70] | 25.6 | 0.724 | - | 43.8 | 0.511 | - | 37.4 | 0.363 | - |
| *Dynamic reconstruction* | | | | | | | | | |
| SurfelWarp [20] | 1.03 | 2.13 | - | 3.75 | 6.53 | - | - | - | - |
| OFlow [59] | 26.0 | 0.732 | 1.69 | 55.2 | 0.412 | 0.812 | 31.5 | 0.461 | 0.907 |
| REDO | **31.9** | **0.647** | **1.47** | 57.4 | 0.349 | **0.765** | **41.6** | **0.337** | **0.846** |

Table 1: **Quantitative results.** For both shape reconstruction (mIoU and mCham.) and dynamics modeling (mACD), REDO demonstrates significant improvements over prior methods. mACD is not available for static methods and SurfelWarp which don't predict temporally corresponding meshes.

1) SAIL-VOS 3D [27]: a photo-realistic synthetic dataset extracted from the game GTA-V. It consists of RGBD videos together with ground-truth (masks and cameras). Out of the original 178 object categories, we use 7 dynamic ones: human, car, truck, motorcycle, bicycle, airplane, and helicopter. During training, we randomly sample clips from 193 training videos. For evaluation, we sample 291 clips from 78 validation videos. We further hold out 2 classes (dog and gorilla) as an unseen test set.

2) DeformingThings4D++: DeformingThings4D [42] is a synthetic dataset containing 39 object categories. As the original dataset only provides texture-less meshes, we render RGBD video and corresponding ground-truth (mask and camera) using Blender. Because the original dataset doesn't provide dataset splits, we create our own. Specifically, during training, we randomly sample clips from 1227 videos. For evaluation, we create a validation set of 152 clips and a test set of 347 clips. We hold out class puma with 56 videos as a zero-shot test set. We dub this dataset DeformingThings4D++.

3) 3D Poses in the Wild (3DPW) [86]: to test the generalizability of our model, we test on this real-world video dataset. Unfortunately, no real-world multi-class 4D dataset is available. Therefore, we test REDO in a class-specific, *i.e.*, class human, setting using 3DPW. This dataset contains calibrated videos, *i.e.*, known camera, and 3D human pose annotation. However, it doesn't provide ground-truth mesh and depth. To extract a mesh, we fit the provided 3D human pose using the SMPL [47] template. To compute depth, we use Consistent Video Depth (CVD) [51] with ground-truth camera data to get temporally consistent estimates. The dataset contains 60 videos (24 training, 12 validation, and 24 testing). During training, we randomly sample clips from all training videos. For evaluation, we evaluate at uniformly sampled clips (10 clips per video) using the validation and test set.

**Baselines.** We consider the following baselines: 1) Static reconstruction: we adopt state-of-the-art methods ONet [53] and PIFuHD [70], and train them for per-frame static reconstruction. For a fair comparison, we train these two networks in a class-agnostic setting using all the frames in the training videos. 2) Fusion-based dynamic reconstruction: most fusion based methods [57, 74, 75] are neither open-sourced nor reproduced. Among the available ones, we adapt the author-released SurfelWarp [20] due to its superior performance. Since this method is non-parametric and requires no training, we directly apply it on the validation/testing clips. 3) Supervised dynamic reconstruction: REDO learns to reconstruct dynamic objects in a supervised manner. OFlow [59] also falls into this category but it doesn't handle the partial observation and rigid-motion. Note, REDO uses each clip's center frame as the canonical space while OFlow uses the initial one. Therefore, for a fair comparison, we set OFlow's first frame to be the center of our input clip.

**Metrics.** To evaluate the reconstructed geometry, we report the mean volumetric Intersection over Union (mIoU) and the mean Chamfer $\ell_1$ distance (mCham.) over different classes at one time-step, *i.e.*, the center frame of the test clip. To evaluate the temporal motion prediction, we compute the mean Averaged (over time) Correspondence $\ell_2$ Distance (mACD) following [59]. As stated before, OFlow's starting frame is set to the center frame. For a fair comparison, we report mACD on the latter half of each testing clip. We compute mCham. and mACD error in a scale-invariant way following [53, 59, 18]: we use 1/10 times the maximal edge length of the object's bounding box as unit one. Even though our network is class-agnostic, we report the mean values over different object categories. Namely, all 'mean' operations are conducted over categories.

### 4.3 Quantitative results

We present results on all three datasets in Tab. 1. For a fair comparison, we test on the center frame (canonical frame of REDO) of the validation/testing clips. We observe that:

1) REDO improves upon the static methods for shape reconstruction on SAIL-VOS 3D and 3DPW (+6.3/4.2 mIoU and -0.077/-0.026 mCham. over best static method). This is because the static methods cannot capture the visual information from other frames in the video clip and thus fail to handle partial visibility. However, REDO performs slightly worse than ONet on DeformingThings4D++ due to the unrealistic simplified visual input: 1) the pictures have only one foreground object with neither occlusion nor background, 2) the rendered color is determined by the vertices order and hence provides a visual short-cut for 2D to 3D mapping. Without modeling dynamics, the static baselines are hence easier to optimize. In contrast, SAIL-VOS 3D renders photo-realistic game scenes with diverse dynamic objects, which are much closer to a real-world setting.

2) REDO outperforms fusion-based method SurfelWarp greatly on all benchmarks, as SurfelWarp only recovers the observable surface rather than the complete shape. We didn't run SurfelWarp on 3DPW as it relies on precise depth as input and crashes frequently using the estimated depth values.

3) REDO improves upon OFlow (+5.9/2.2/10.1 mIoU and -0.085/0.063/0.124 mCham.) for shape reconstruction due to the pixel-aligned 4D implicit representation, whereas OFlow encodes the whole image as a single feature vector and looses spatial information.

4) Regarding dynamics modeling, REDO improves upon OFlow (-0.22/0.047/0.061 mACD) thanks to the pixel-aligned implicit flow-field. Note that OFlow normalizes the 3D models at each time-step into the first frame's model space and hence fails to capture rigid motion like translation. In contrast, our canonical space is constructed for the entire clip in which REDO predicts a complete trajectory.

As stated in Sec. 3.3, the reconstructed mesh on the center frame is propagated to other frames in the video clip for a dense reconstruction. We thus report the per-frame results in Appendix § E. In addition, all above results are mean values averaged over object categories to avoid being biased towards the most frequent class. Class-wise results on SAIL-VOS 3D are reported in Appendix § C.

**Zero-shot reconstruction.** To test the generalizability of REDO, we further test on unseen categories with no fine-tuning in Tab. 2. The result is averaged over three unseen classes: dog and gorilla from SAIL-VOS 3D, and puma from DeformingThings4D++. REDO still greatly outperforms baselines and doesn't fail catastrophically. The per-class results are provided in Appendix Tab. S5.

|  | mIoU↑ | mCham.↓ | mACD↓ |
|---|---|---|---|
| ONet | 23.1 | 0.764 | - |
| PIFu | 21.2 | 0.911 | - |
| SurfelWarp | 2.06 | 1.23 | - |
| OFlow | 26.7 | 0.931 | 1.18 |
| REDO | **38.5** | **0.479** | **1.07** |

Table 2: **Zero-shot reconstruction.**

### 4.4 Analysis

In Tab. 3, we provide an ablation study of different components in REDO using SAIL-VOS 3D data. 1) We first replace the temporal aggregator $f_{agg}$ with an average pooling layer where features of different frames are averaged and fed into the shape reconstruction and velocity field network. The results are shown in the 1$^{st}$ row of Tab. 3 (avg. pooling). The performance drops by -3.6 IoU, +0.065 mCham., and +0.13 mACD. 2) We then study

|  | mIoU↑ | mCham.↓ | mACD↓ |
|---|---|---|---|
| avg. pooling | 28.3 | 0.712 | 1.60 |
| w/o alignment | 24.1 | 0.937 | 1.85 |
| w/o $\mathcal{L}_{temp}$ | 29.4 | 0.685 | 3.12 |
| REDO | **31.9** | **0.647** | **1.47** |

Table 3: **Ablation studies.**

pixel-aligned feature representations $\mathbf{x_p}, \mathbf{z_p}$. We replace these two features with the feature map of the entire input frame following OFlow [59] but still keep the transformer to aggregate these feature maps. Results of this ablation are reported in Tab. 3 (w/o alignment). Compared to REDO, this setting greatly hurts the results (-7.8 mIoU, +0.290 mCham., -0.38 mACD) as the network can no longer handle partial observations and the 3D predictions don't well align with the visual input. 3) In many real-world tasks, ground-truth meshes of different time-steps are not corresponded. Conceptually, REDO could adapt to this setting. This is because all components are differentiable and the flow-field network could be used as a latent module for shape reconstruction. To mimic this setting, we train REDO using only the shape reconstruction loss $\mathcal{L}_{shape}$. As shown in Tab. 3 (w/o $\mathcal{L}_{temp}$), the model still recovers objects at the canonical frame. However, mACD increases significantly (+1.65).

### 4.5 Qualitative results

Fig. 4 shows a few representative examples of REDO predictions on SAIL-VOS 3D and DeformingThings4D++. Please check Appendix § F for more results on real-world data and additional analysis. From Fig. 4 we observe that: 1) REDO is able to recover accurate geometry and dynamics of different objects from input video frames. It completes the occluded parts and hallucinates invisible

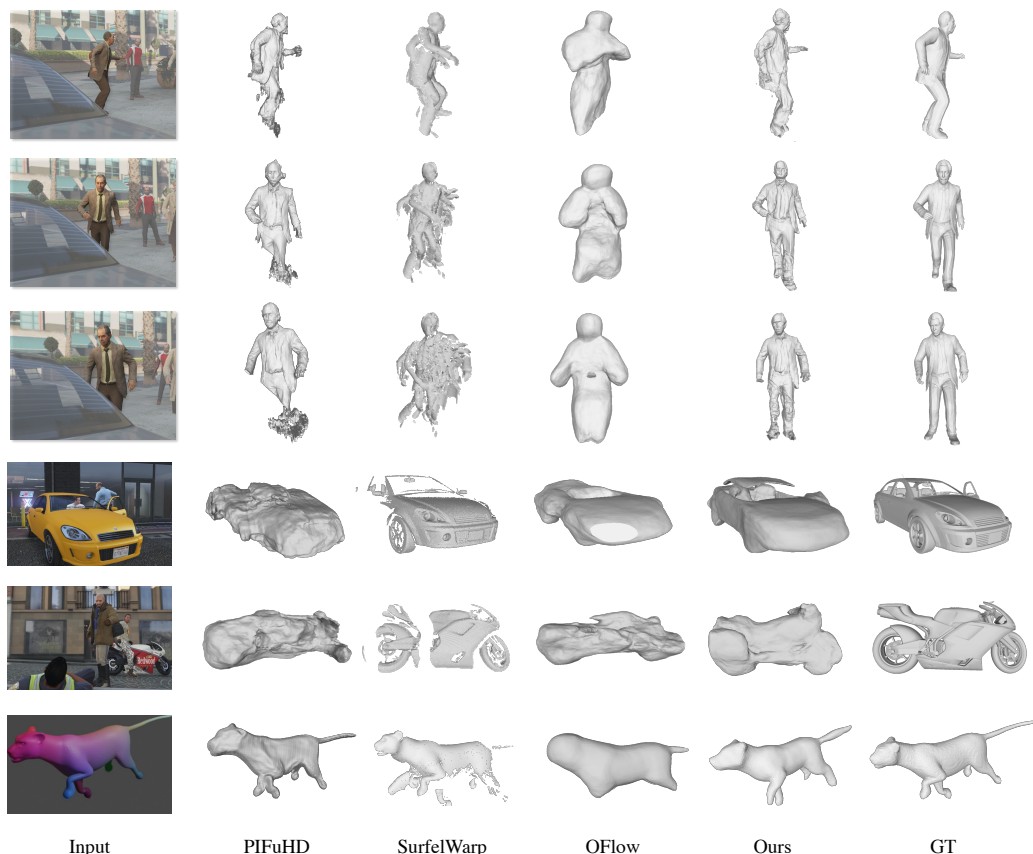

| Input | PIFuHD | SurfelWarp | OFlow | Ours | GT |
| --- | --- | --- | --- | --- | --- |

Figure 4: **Qualitative results.** We illustrate the input frames with the object of interest highlighted, the reconstructed meshes obtained from different methods, and the ground-truth mesh.

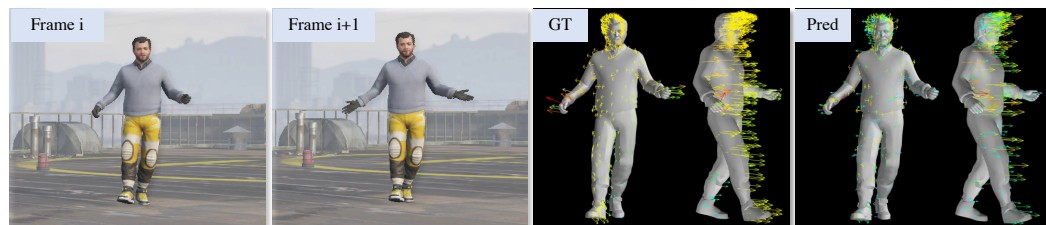

Figure 5: **Flow-field visualization.** REDO accurately recovers the non-rigid motion (*e.g.*, moving forward) but is less precise for small-scale articulation (*e.g.*, hand).

components (*e.g.*, legs and back of humans; rear tire of the motorcycle) by aggregating temporal information and due to large-scale training. 2) REDO improves upon baselines methods. *E.g.*, PIFuHD struggles to handle occlusion and non-human objects, SurfelWarp only predicts the visible surface, and OFlow results are over-smooth as it ignores the spatial information. 3) REDO predictions are still far from perfect compared to ground-truth meshes. Many fine-grained details are missing (*e.g.*, clothing of the human, car's front-light and tires, *etc*.).

We also visualize the predicted and ground-truth motion vectors in Fig. 5. REDO successfully models the rigid motion, *i.e.*, moving forward, over the entire human body, *e.g.*, head, leg, chest, *etc*. It's less accurate in capturing the very fine-grained dynamics, *e.g.*, the hand motion where the ground-truth indicates the hand will open while the prediction doesn't.

## 5   Conclusion

We present REDO, a novel class-agnostic method to reconstruct dynamic objects from videos. REDO is implemented as a canonical 4D implicit function which captures the precise shape and dynamics and deals with partial visibility. We validate the effectiveness of REDO on synthetic and real-world datasets. We think REDO could be generalized to a wide variety of 4D reconstruction tasks.

**Acknowledgements & funding transparency statement.** This work was supported in part by NSF under Grant #1718221, 2008387, 2045586, 2106825, MRI #1725729, NIFA award 2020-67021-32799 and Cisco Systems Inc. (Gift Award CG 1377144 - thanks for access to Arcetri). ZR is supported by a Yee Memorial Fund Fellowship.

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
