# Appendix:
# Class-agnostic Reconstruction of Dynamic Objects from Videos

In this appendix we first provide additional implementation details (§ A) before providing more information about the datasets (§ B). We then discuss additional quantitative results including per-class reconstruction results (§ C), the propagated per-frame reconstruction results (§ E), and per-class zero-shot reconstruction results (§ D). Lastly, we illustrate additional qualitative results (§ F).

## A    Implementation Details

We provide additional implementation details in this section.

**Baselines.**  For the baseline models, we used the released code of ONet [53], OFlow [59] and PIFuHD [69, 70]. For the static 3D reconstruction baselines (*e.g*., ONet and PIFuHD), we re-train these models on all the frames from the training set using the default setting. Note, the dynamic 4D reconstruction baseline OFlow always utilizes the initial frame for the canonical space. Therefore, for a fair comparison, we set OFlow's first frame to be the center of our input clip.

**ODE.** To solve the Ordinary Differential Equation (ODE), we use the Dormand–Prince method with a relative error tolerance of $1e^{-2}$ and an absolute error tolerance of $1e^{-4}$. To support batch-wise processing in the ODE solver for the backward flow, we use the variant from OFlow [59]. We also zero-pad the velocity network output to make it compatible with a ODE solver following OFlow [59].

**Training.** Our code is implemented in PyTorch. The training process takes around 30 hours on a 4 GPU machine. To ensure a fair comparison, parts of our model, *i.e*., the image encoder and the reconstruction net, share the same architecture as PIFuHD.

## B    Datasets

We provide a detailed comparison of popular dynamic 3D dataset in Tab. S1. Among them, we choose to use SAIL-VOS 3D, DeformingThings4D, and 3D Poses in the Wild (3DPW) because of the dataset scale and the quality of the ground-truth labels. Besides SAIL-VOS 3D, the other two are incomplete and miss some ground-truth labels. In below section, we introduce the details about how we complete the datasets used in our experiments.

| Datasets | Type | #Category | #Videos | RGB | Depth | GT camera | GT mask | GT mesh | Misc. GT |
|---|---|---|---|---|---|---|---|---|---|
| *Single-class real-world datasets* | | | | | | | | | |
| Human3.6M [29] | real-world | 1 (human) | N/A | ✓ | ✓ | ✓ | ✗ | ✗ | |
| 3DPW [87] | real-world | 1 (human) | 60 | ✓ | ✗ | ✓ | ✗ | ✗ | 3D pose |
| D-FAUST [3] | real-world | 1 (human) | N/A | ✗ | ✗ | ✗ | ✗ | ✓ | |
| *Multi-class real-world datasets* | | | | | | | | | |
| KillingFusion [74] | real-world | 5 | 5 | ✓ | ✓ | ✓ | ✓ | ✗ | |
| VolumeDeform [28] | real-world | 8 | 8 | ✓ | ✓ | ✓ | ✗ | ✓ | |
| DeepDeform [5] | real-world | multiple | 400 | ✓ | ✓ | ✓ | ✓ | ✗ | |
| *Multi-class synthetic datasets* | | | | | | | | | |
| DeformingThings4D++ [42] | synthetic | 39 | 1972 | ✗ | ✗ | ✗ | ✗ | ✓ | |
| SAIL-VOS 3D [27] | synthetic | 10 | 484 | ✓ | ✓ | ✓ | ✓ | ✓ | |

Table S1: **Summary of dataset statistics.**

### B.1    SAIL-VOS 3D

SAIL-VOS 3D [27] is a very challenging dataset due to the diverse appearance and complex motion. Most observed objects are partially visible and the occlusion is often changing across time. The dataset also has a very challenging long-tail distribution.

**Data source.** SAIL-VOS 3D is publicly available for research and educational purpose under a user agreement.[2] This dataset is extracted from the photo-realistic game Grand Theft Auto (GTA) V.[3] The collected data of SAIL-VOS 3D contains neither personally identifiable information nor offensive content.

---

[2] http://sailvos.web.illinois.edu/_site/index.html
[3] https://www.rockstargames.com/games/V

| Split | #Videos | #Frames | PO | Size | #Clips | Class-wise distribution |
|---|---|---|---|---|---|---|
| Train | 193 | 475 | 84% | 15% | - | #**classes = 7**: person (188), car (62), truck (10), helicopter (7), motorcycle (10), bicycle (3), airplane (3) |
| Val | 78 | 675 | 86% | 13% | 291 | #**classes = 8**: person (61-241), car (12-23), motorcycle (4-8), truck (2-7), bicycle (2-5), trolley (2-3), airplane (1-2), helicopter (1-2) |
| Unseen | 2 | 333 | 100% | 6.4% | 2 | #**classes = 2**: dog (1-1), gorilla (1-1) |

Table S2: **SAIL-VOD 3D statistics.** Values in parenthesis $(V - C)$ indicate the number of videos $V$ and clips $C$ for the corresponding class. For each split we report the number of videos (#Videos), average video frames (#Frames), partial visibility ratio (PO), ratio between object area and image area (Size), and the number of sampled clips (#Clips). Note, 1) videos are untrimmed and may contain several objects from different classes. Therefore, #Videos may not equal the summation of $V$ from all classes; 2) #Clips for split train is not reported as we randomly sample clips from videos during training.

**Splits & statistics.** Our network is class-agnostic and handles different dynamic objects in a unified manner. For this purpose, we use 10 kinds of dynamic objects, *i.e.*, human, car, truck, motorcycle, bicycle, airplane, helicopter, gorilla, trolley, and dog. We provide detailed statistics in Tab. S2

**Clips sampling.** SAIL-VOS 3D contains videos whose number of frames ranges from single digits to several hundreds. For our experiments, we sample fixed-length video clips satisfying the following rules: 1) the average occlusion ratio of the observed object is lower than 0.75 following [27]. 2) the average visible instance area of the object is bigger than $128 \times 128$ pixels. In our training, we set the number of frames to be 17 following [59]. If the length of the original video clip is shorter than 17, we duplicate beginning and ending frames to meet this requirement.

**Ground-truth.** To compute the ground-truth occupancy label, we first truncate the ground-truth mesh of each frame in the input clip to remove the vertices and faces that are never seen in this clip. Note, the set of invisible vertices and faces can be easily identified since SAIL-VOS 3D provides the vertices' temporal correspondences across frames. Therefore, we just need to compute a per-frame set of invisible vertices and faces and then take a union of those per-frame sets. We can then compute the ground-truth occupancy label and correspondences using the converted truncated mesh.

## B.2 DeformingThings4D++

**Data source.** The original dataset DeformingThings4D [42] is released for non-commercial research and educational purposes under "DeformingThings4D Terms of Use"[4], and the accompany code is release under a non-commercial creative commons license. The characters in this dataset are obtained from Adobe Mixamo[5]. This dataset is synthetic and contains neither personally identifiable information nor offensive content.

**Rendering.** The original DeformingThings4D [42] data is composed of 1972 scenes with 1772 animals and 200 humanoids from 39 classes. Note, the number of classes isn't 31 as listed in the paper [42]. This is because the dataset does not contain a mapping between scene IDs and categories. Therefore, we count classes ourselves and treat each unique object type as a class. See Tab. S3 for more details. Each scene in the dataset contains a triangular mesh and vertices's 3D coordinates at different time steps, which provide ground truth motion fields. For each scene, we render with resolution of $500 \times 600$ and obtain the following information with several representative examples shown in Fig. S1.

• Camera: we do not consider camera shot changes and fix the camera pose for the whole animation. Specifically, we first render with a fixed camera pose. If the resulting animations are of low-quality, *e.g.*, too zoomed-in or zoomed-out, we manually adjust to a better pose.

• RGB: in order to maintain correspondences between rendered frames, *i.e.*, same position of an object should have same color across the whole animation, we assign an RGB value to each

---

[4] https://docs.google.com/forms/d/e/1FAIpQLSckMLPBO8HB8gJsIXFQHtYVQaTPTdd-rZQzyr9LIIkHA515Sg/viewform
[5] https://mixamo.com/

| Split | #Videos | #Frames | PO | Size | #Clips | Class-wise distribution |
|---|---|---|---|---|---|---|
| Train | 1227 | 61 | 88% | 8.5% | - | #**classes = 37**: bear (195), deer (170), fox (126), humanoids (111), moose (105), rabbit (72), doggie (45), dragon (41), elk (41), tiger (40), procy (38), raccoon (37), bunny (28), bucks (25), grizz (24), canie (22), huskydog (21), bull (13), milkcow (12), cattle (11), hog (7), rhino (7), cetacea (6), elephant (6), chicken (5), hippo (3), lioness (3), sheep (3), raven (2), cat (1), crocodile (1), duck (1), goat (1), leopard (1), pig (1), seabird (1), zebra (1) |
| Val | 152 | 64 | 96% | 6.9% | 152 | #**classes = 25**: bear (27-27), deer (24-24), fox (18-18), moose (15-15), rabbit (10-10), humanoids (7-7), doggie (6-6), dragon (5-5), elk (5-5), procy (5-5), raccoon (5-5), tiger (5-5), bunny (4-4), bucks (3-3), canie (2-2), huskydog (2-2), bull (1-1), cattle (1-1), cetacea (1-1), chicken (1-1), elephant (1-1), hog (1-1), lioness (1-1), milkcow (1-1), sheep (1-1) |
| Test | 347 | 65 | 94% | 7.2% | 347 | #**classes = 29**: bear (55-55), deer (48-48), humanoids (44-44), fox (36-36), moose (30-30), rabbit (20-20), doggie (12-12), dragon (11-11), elk (11-11), tiger (11-11), procy (10-10), raccoon (10-10), bunny (8-8), bucks (6-6), grizz (6-6), canie (5-5), huskydog (5-5), bull (3-3), cattle (3-3), milkcow (3-3), hog (2-2), cetacea (1-1), chicken (1-1), duck (1-1), elephant (1-1), goat (1-1), lioness (1-1), raven (1-1), sheep (1-1) |
| Unseen | 56 | 65 | 92% | 7.4% | 56 | #**classes = 1**: puma (56-56) |

Table S3: **DeformingThings4D++ statistics.** Values in parenthesis $(V - C)$ indicate the number of videos $(V)$ and clips $(C)$ for the corresponding class. For each split we report the number of videos (#Videos), average video frames (#Frames), partial visibility ratio (PO), ratio between object area and image area (Size), and the number of sampled clips (#Clips). Note, #Clips for split train is not reported as we randomly sample clips from videos during training.

vertex on the mesh and keep that RGB value for all time steps. We then utilize Blender's `ShaderNodeVertexColor` to get RGB frames.

• Mask & depth: we ray-cast from each pixel of a frame. 1) For mask, we mark a pixel as "in mask" if its ray reaches the object's mesh. 2) For depth, if the ray reaches the mesh, we store the $z$ coordinate of the intersection point between ray and mesh in the camera coordinate system.

**Splits & Statistics.** Note, the original dataset doesn't provide the train/validation/test splits. We hence create our own split following the rules discussed next: First, we discard the videos with invalid motion values, *e.g.*, `NaN`. Second, we do not use videos if there are frames where objects leave the view frustum entirely. After filtering, we split the remaining videos into train/validation/test sets with a rough ratio of 7 : 1 : 2: 1) We reserve one class for zero-shot experiments to examine algorithm generalization. Specifically, we remove the class puma from the training/validation/test set as an unseen category. 2) For validation and test sets, we consider a subset of videos that composes of at least one rendered frame where the object is completely visible. The motivation is that if an object appears completely, we do not need to impose any extra processing for evaluation, *e.g.*, truncating the ground truth mesh. In this way, we avoid inaccuracies incurred in the processing step. We sample

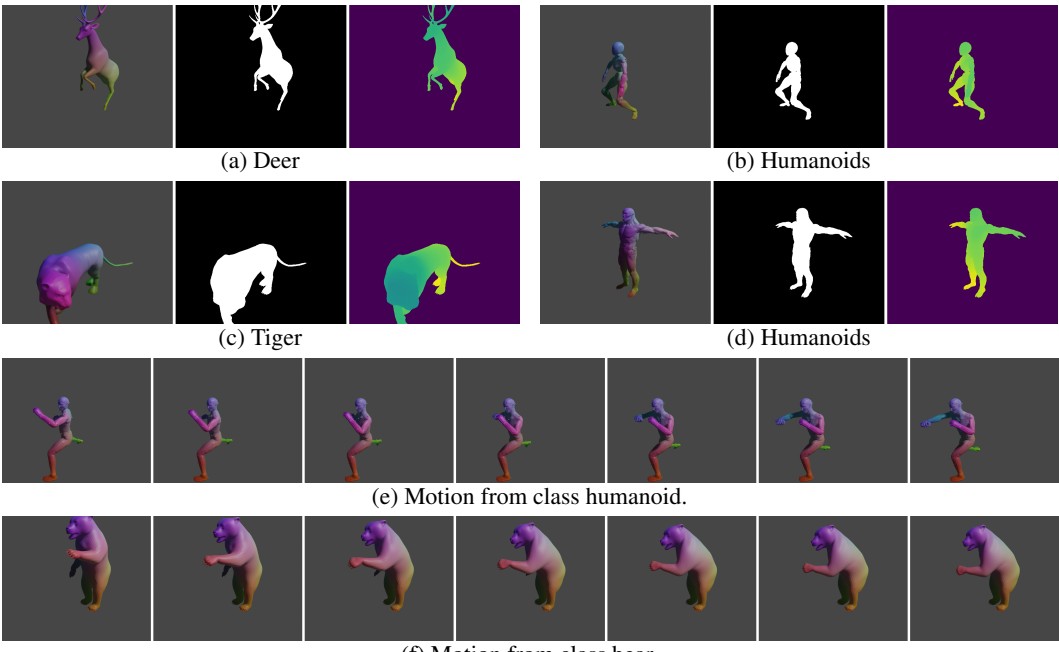

(a) Deer            (b) Humanoids

(c) Tiger            (d) Humanoids

(e) Motion from class humanoid.

(f) Motion from class bear.

Figure S1: **Examples of DeformingThings++. (a) - (d):** we display the rendered RGB image (left), mask (center), and depth (right) of 4 instances from different objects; **(e) - (f):** we showcase dynamics of two objects. Note the rendered images maintain semantic consistency across frames.

152 and 347 videos from the subset for validation and test respectively. 3) For the remaining videos, we use them as the training set. Please check Tab. S3 for detailed split distribution and statistics.

### B.3   3D Poses in the Wild (3DPW)

3DPW is a real-world 3D video dataset focusing on humans. The collected videos usually contain multiple people, which are often occluded and out of the camera view. The camera is also moving to capture the human dynamics. Due to all these appealing and challenging properties, we adopt it in our experiments to test the generalizability of REDO under a class-specific setting.

**Data source.** 3DPW is publicly available under a license.[6] This dataset captures real-world human videos with a moving camera and Inertial Measurement Units (IMUs) in complex scenes. This dataset contains no offensive content.

**Ground-truth.** For training and evaluation purposes, we estimate the human mesh. Since 3D pose annotation is given, we utilize the Skinned Multi-Person Linear (SMPL) [47] human template. Since the same template is used across all frames in a video, we also get the ground-truth correspondences (vertices). Note that we render an amodal mesh in this experiment under the class-specific setting, meaning we aim to reconstruct the complete human mesh when the visual inputs are partially visible.

## C   Per-class results

To further analyze the performance of REDO, we provide class-specific results on SAIL-VOS 3D in Tab. S4. Overall, REDO outperforms baseline methods in 18 out of 21 metrics. In terms of performance across classes, REDO performs best for human and car because these two classes have the most training examples. REDO is less effective for airplane and bicycle due to the complex geometry and the small number of training examples.

---

[6] https://virtualhumans.mpi-inf.mpg.de/3DPW/license.html

| Classes | human | | | car | | | truck | | | airplane | | |
|---|---|---|---|---|---|---|---|---|---|---|---|---|
| Metrics | mIoU↑ | mCham.↓ | mACD↓ | mIoU↑ | mCham.↓ | mACD↓ | mIoU↑ | mCham.↓ | mACD↓ | mIoU↑ | mCham.↓ | mACD↓ |
| ONet [53] | 39.2 | 0.996 | - | 36.6 | 0.770 | - | 24.5 | 0.942 | - | 15.7 | 1.22 | - |
| PIFuHD [70] | 42.7 | **0.646** | - | 28.6 | 0.726 | - | 23.2 | 0.751 | - | 15.4 | 0.802 | |
| SurfelWarp [20] | 0.841 | 2.20 | - | 0.609 | 3.52 | - | 0.171 | 1.72 | - | 1.54 | 1.51 | |
| Oflow [59] | 37.3 | 0.748 | 1.87 | 35.6 | 0.858 | 1.71 | 21.6 | 0.801 | 1.94 | 21.9 | **0.588** | |
| REDO | **45.2** | 0.657 | **1.54** | **40.1** | **0.672** | **1.66** | **30.3** | **0.709** | **1.70** | **24.6** | 0.615 | **0.904** |

| Classes | helicopter | | | bicycle | | | motorcycle | | |
|---|---|---|---|---|---|---|---|---|---|
| Metrics | mIoU↑ | mCham.↓ | mACD↓ | mIoU↑ | mCham.↓ | mACD↓ | mIoU↑ | mCham.↓ | mACD↓ |
| ONet [53] | 22.6 | 0.986 | - | 13.5 | 0.898 | - | 19.3 | 0.846 | - |
| PIFuHD [70] | 25.4 | 0.725 | - | 20.4 | **0.718** | - | 23.3 | 0.701 | - |
| SurfelWarp [20] | 1.42 | 3.74 | - | 1.50 | 5.81 | - | 1.09 | 1.10 | - |
| Oflow [59] | 24.8 | 0.526 | 1.12 | 18.7 | 0.831 | 2.49 | 22.3 | 0.770 | 1.72 |
| REDO | **30.9** | **0.504** | **1.01** | **23.4** | 0.730 | **1.95** | **28.5** | **0.639** | **1.53** |

Table S4: **Per-class reconstruction results on SAIL-VOS 3D.**

# D  Zero-shot reconstruction

To demonstrate REDO's class-agnostic characteristics and generalization ability, we evaluate REDO on held-out classes. Specifically, we test on classes dog and gorilla from SAIL-VOS 3D and puma from DeformingThings4D++ in Tab. S5. REDO outperforms baselines for unseen classes dog and gorilla on synthetic dataset SAIL-VOS 3D and is competitive on puma from DeformingThings4D++. For class puma, SurfelWarp produces really low IoU while maintaining low Chamfer distance. The reason: although SurfelWarp only reconstructs observable parts as stated in Sec. 4.3, we find the reconstructed geometry overlaps well with the ground truth on this class. Therefore, many vertices on the mesh have near-zero Chamfer distances, resulting in a 0.302 final result in Tab. S5.

| Dataset | SAIL-VOS 3D | | | SAIL-VOS 3D | | | DeformingThings4D++ | | |
|---|---|---|---|---|---|---|---|---|---|
| Class | dog | | | gorilla | | | puma | | |
| Metrics | mIoU↑ | mCham.↓ | mACD↓ | mIoU↑ | mCham.↓ | mACD↓ | mIoU↑ | mCham.↓ | mACD↓ |
| ONet [53] | 23.1 | 0.727 | - | 19.5 | 0.866 | - | 48.5 | 0.322 | - |
| PIFuHD [70] | 19.3 | 0.928 | - | 31.2 | 0.516 | - | 46.4 | 0.389 | - |
| SurfelWarp [20] | 0.893 | 1.46 | - | 0.984 | 1.93 | - | 4.30 | 0.302 | - |
| Oflow [59] | 33.1 | 0.627 | 1.51 | 25.4 | 0.950 | 1.37 | 47.4 | 0.407 | 0.661 |
| REDO | 36.2 | 0.597 | 1.24 | 34.5 | 0.403 | 1.32 | 44.9 | 0.436 | 0.679 |

Table S5: **Per-class zero-shot reconstruction results.**

# E  Per-frame results

Recall that REDO reconstructs the mesh in the canonical space and then propagates the reconstructed geometry to every frame in the video clip. To further analyze the performance across frames, we provide dense reconstruction results on SAIL-VOS 3D in Fig. S2. Note, REDO uses each clip's center frame as its canonical space while OFlow uses the 1st frame. Therefore, we show REDO performance for all frames and OFlow for frames from the middle to the last one. As can be seen clearly, REDO outperforms OFlow on all three metrics across frames. However, the performance of both methods drops quickly as the reconstruction is propagated to frames far from the canonical one. This suggests that a better flow-field network is required, which could be an interesting future research direction.

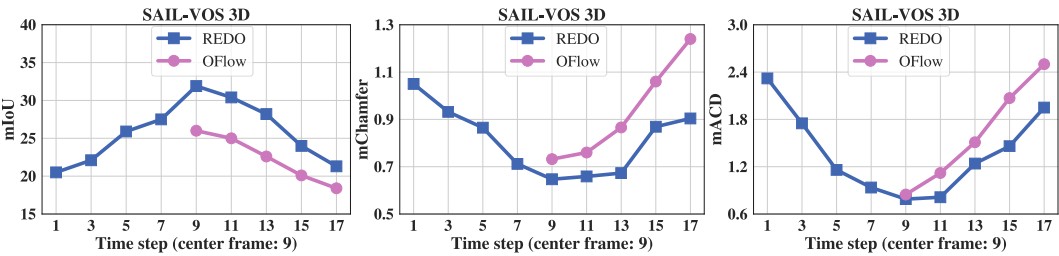

Figure S2: **Per-frame results.** We show mIoU, mCham., mACD from left to right.

# F Additional qualitative results

In Fig. S3, we show additional qualitative comparisons on the real-world 3DPW dataset. In each subfigure we show the input frame, the estimated mask of the object of interest using Mask R-CNN [26], the prediction of the best-performing baseline model (PIFuHD), and our prediction. From Fig. S3 we observe that REDO recovers complete and smooth prediction of the real-world human in various poses. In contrast, the baseline method often struggles when humans are in rare poses. This is largely due to the difference in the pre-training dataset where PIFuHD is trained on stand-still humans with no occlusion, whereas REDO is trained on the large-scale synthetic dataset SAIL-VOS 3D which comes with humans in challenging poses and with sever occlusions.

In Fig. S4, we demonstrate our prediction on SAIL-VOS 3D from multi-views in a single video clip. In each chunk we show the input frames, ground truth mesh, and predicted mesh of one video clip from top to bottom. From Fig. S4 we can see that the front-view of REDO's prediction aligns well with input images. Meanwhile, REDO also hallucinates decently about the side (partially observed) and back (invisible) of the objects. The predicted meshes are relatively smooth across time.

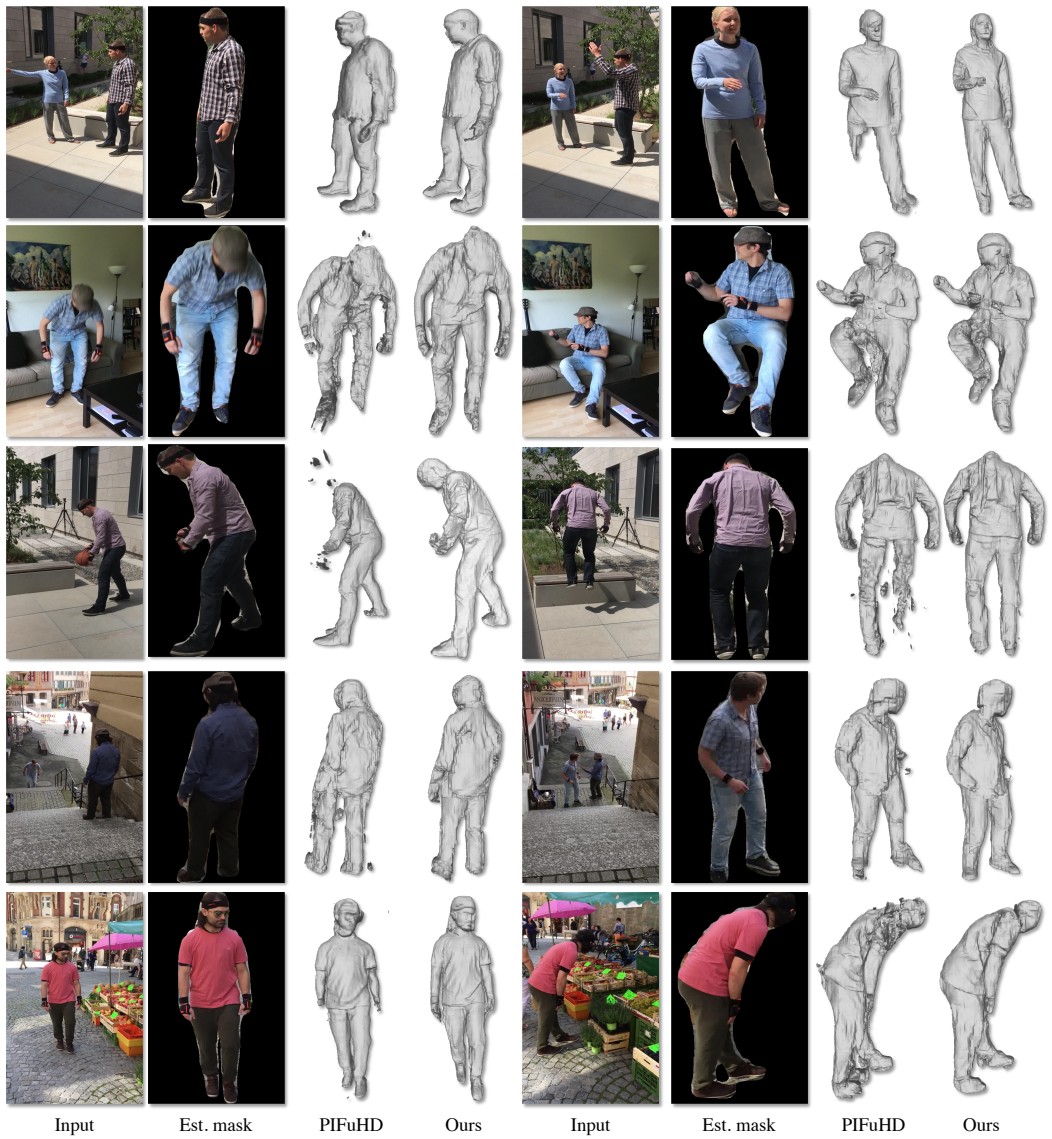

Input     Est. mask     PIFuHD     Ours        Input     Est. mask     PIFuHD     Ours

Figure S3: **Qualitative results on 3DPW.** From left to right, we visualize the input image, the estimated mask of the object of interest, the prediction of the best-performing baseline model (PIFuHD), and our prediction.

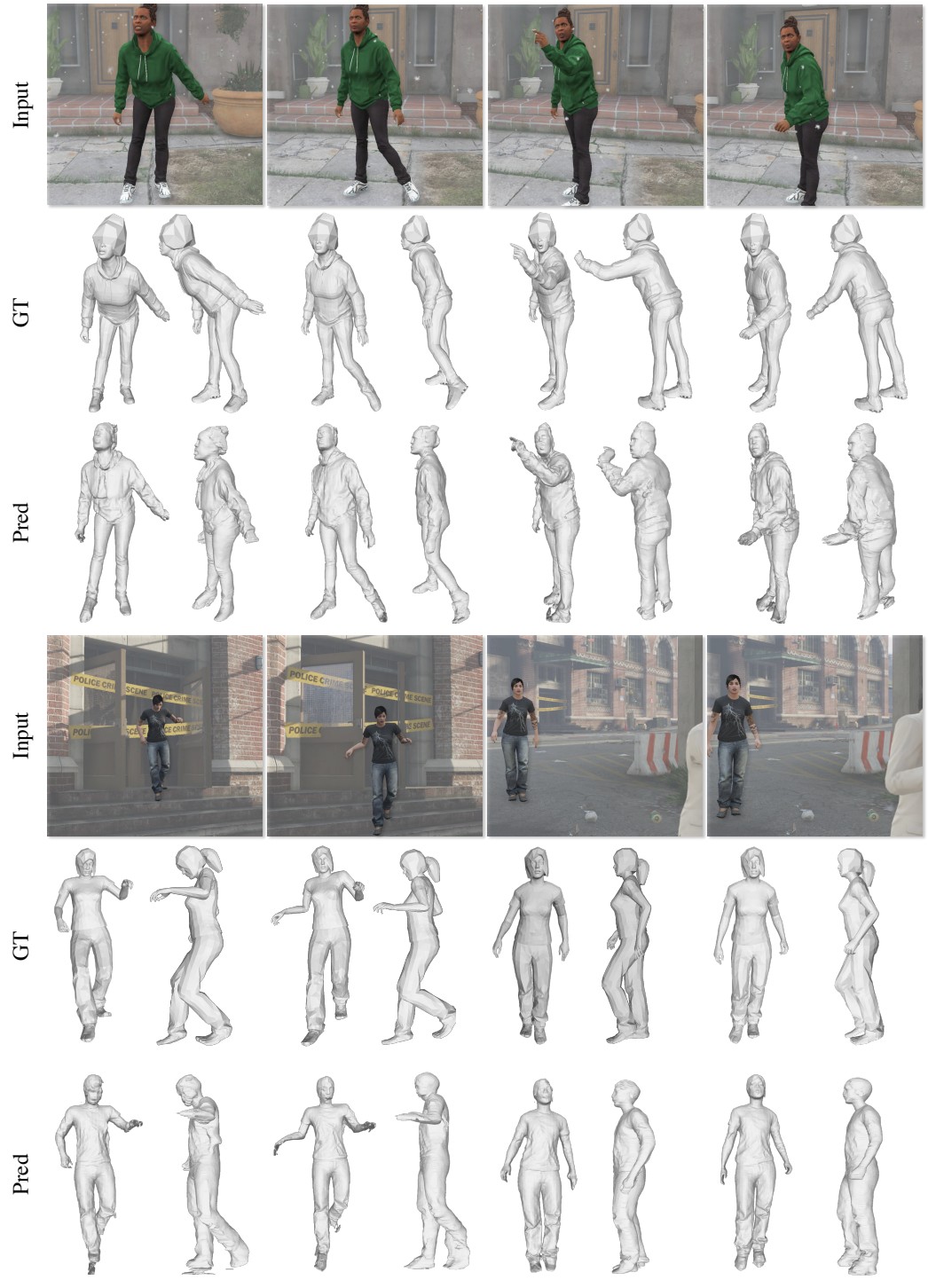

Figure S4: **Qualitative results.** We illustrate the input frames with the object of interest highlighted, the ground-truth mesh, and the reconstructed mesh (front and side views).