# OpenReview forum: "Class-agnostic Reconstruction of Dynamic Objects from Videos"
_NeurIPS.cc/2021/Conference — NeurIPS 2021 Poster_

### Official Review · Reviewer_nToe · 2021-07-16

**Rating:** 4
**Confidence:** 4

**Summary:**

This paper presents an approach for 4D (space + time) reconstruction from RGBD video sequences with instance masks. It uses an implicit function based representation with explicit modeling of velocity fields (similar to Occupancy Flow) to model dynamic objects. demonstrates superior performance on the SAIL VOS 3D dataset compared to prior works. The authors claim their key contributions to be 1) a pixel-aligned neural representation for non-rigid dynamic objects; and (2) an end-to-end trainable framework (FREDO) for learning this representation from monocular video clips showing partially observed objects.

**Limitations And Societal Impact:**

- I believe the limitations of this work around the nature of supervision required and comparisons of inputs required vs baselines is not discussed adequately. A detailed discussion around the baselines would significantly improve the manuscript.

- The authors are also encouraged to comment on the novelty of this method in the context of works like PiFU and OFlow. Specifically, what are the technical contributions of this work beyond combining ideas from these two works?

**Main Review:**

Strengths
- The paper combines two key recent ideas which have been shown to be very effective for representing 3D / 4D objects - pixel aligned representations (PiFU / PiFUHD) and occupancy flow. The benchmarks show clear improvement over these works in isolation.

- The paper is clearly written and easy to understand. The notation is clear and abstractions are used when necessary to improve clarity (e.g. avoiding basic 3D<->2D and 3D <->3D transform equations).

- The illustrations in the paper are very helpful and make understanding the work much easier. Fig 4 and the associated discussion regarding quality of flow fields recovered is also quite interesting.

Weaknesses
- I am concerned that the novelty of the work is limited by the fact that it is a straightforward combination of pixel aligned implicit functions and occupancy flow. The contribution of dealing with partially observed instances is rather weak as it involves just aggregating point clouds over a few frames and constructing the canonical space.

- The supervision used by the method is very extensive. Not only does it require RGBD inputs with instance segmentations during inference, it also requires temporal 3D correspondences (scene flow) during training as well. This type of supervision is only possible in synthetic datasets and raises questions on the general applicability of the method.

- I am also concerned about the baselines reported in this paper and how fair the comparisons are. Based on my knowledge of PiFU HD, it does not require depth maps as input during inference while this work does. Similarly, its unclear what the input is for the OFlow baseline. In general, the experiments section lacks details about the baselines (what data they were retrained on, what was the input etc.).

**Time Spent Reviewing:**

2

---

> ### Author Response · Authors · 2021-08-10
> **Response to Reviewer nToe**
>
> ### [Weakness 1] Straightforward combination of Oflow and PIFuHD
>
> FREDO is not a simple combination of Oflow and PIFuHD. The key differences compared to Oflow are: 1) we propose a canonical space representation to tackle object’s occlusion (Sec. 3.1) that Oflow does not take into account; 2) we utilize image features to provide more information about temporal coherence, e.g., discussion in L191-192 about $\mathbf{z}_\mathbf{p}$. Without such information, performance decreases from 32.5 to 26.4 (IoU) (L324-328); 3) we use a transformer to aggregate information from sequences (L164). This design improves IoU from 29.7 to 32.5 compared to vanilla average pooling (L316-323).
>
> Compared to PIFuHD, FREDO 1) develops a canonical space representation; 2) we consider dynamic objects instead of static ones.
>
> ---
>
> ### [Weakness 1] Contribution of dealing with partially observed instances
>
> We respectfully disagree that this contribution is “rather weak as it involves just aggregating point clouds over a few frames and constructing the canonical space”. Besides canonical space, we also develop a transformer-based temporal aggregator (Sec. 3.2.a) to integrate information across frames. Without such aggregation, the performance drops from 32.5 IoU, 1.27 Chamfer distance, and 1.19 L2 correspondence error to 29.7 IoU, 1.54 Chamfer distance, and 1.30 L2 correspondence error with naive average pooling (Tab. 2, L316-323).
>
> ---
>
> ### [Weakness 2] Supervision of GT mesh and flow during training is too strong
>
> Three trends led to the decision to study the use of ground truth meshes and flows for the challenging task of non-rigid dynamic object-agnostic mesh reconstruction. 1) Supervised learning has been shown again and again to be remarkably successful. In fact, most 3D reconstruction works (e.g., ONet [26], PIFu [36], etc.) are trained using GT synthetic meshes and then generalized to real-world data. 2) Recent advances have shown the benefits of large-scale synthetic data, e.g., SunCG [a], ShapeNet [b], Playing for Benchmark [c], etc. 3) The ability of transfer learning has also been verified in recent synthetic platforms, e.g., embodied AI [d], optical flow [e], and autonomous driving [f] etc.
>
> Moreover, large-scale real world data for non-rigid dynamic object-agnostic mesh reconstruction isn’t available. Instead, people either apply costly motion capture systems to limited dynamic object categories [g] or utilize 2D supervision and only evaluate on 2D metrics [h]. Instead, we conduct experiments on large-scale 3D datasets and evaluate the results rigorously using perfect pixel-aligned 3D GT.
>
> [a] Song et al., Semantic Scene Completion from a Single Depth Image. CVPR 2017.
>
> [b] Chang et al., ShapeNet: An Information-Rich 3D Model Repository. CVPR 2015.
>
> [c] Richter et al., Playing for Benchmarks. ICCV 2017.
>
> [d] Savva et al., Habitat: A Platform for Embodied AI Research. ICCV 2019.
>
> [e] Butler et al., A Naturalistic Open Source Movie for Optical Flow Evaluation. ECCV 2012.
>
> [f] Dosovitskiy et al., CARLA: An Open Urban Driving Simulator. CoRL 2017.
>
> [g] Slavcheva et al., KillingFusion: Non-rigid 3D Reconstruction without Correspondences. CVPR 2017.
>
> [h] Yang et al., LASR: Learning Articulated Shape Reconstruction from a Monocular Video. CVPR 2021.
>
> ---
>
>
> ### [Weakness 2] Supervision of GT mask and depth is too strong during inference
>
> We agree that the use of GT information as input is a strong assumption. However, we think use of ground truth is insightful to the community when developing an end-to-end trainable framework for recovering a pixel-aligned representation of non-rigid dynamic objects which may only be partially visible within a frame. By ignoring input noise and by splitting development into manageable pieces, we can reveal the importance of individual parts. For instance, 1) accurate camera data isolates the development of temporal propagation and fusion (Sec. 3.2a) and a pixel-aligned representation (Sec. 3.2b) from other factors; 2) GT depth and mask let us focus on developing a canonical 4D implicit function (Sec. 3.1). Through experiments, we validated the effectiveness of individual modules for the task of 4D reconstruction (Sec. 4.4).
>
> In practice, these assumptions can be relaxed for real-world data. To demonstrate the robustness of FREDO to non-GT inputs, we evaluate FREDO on a real world dataset which only contains RGB sequences. Please see the general responses for more details.
>
> Please see the general response part for results on real world dataset.
>
> ---
>
> ### [Weakness 3] Clarification about baselines
>
> We train PIFuHD with GT RGBD input. Since PIFuHD normalizes all depth values into a predefined range, it does not require depth input during inference. Therefore, during inference, we only use RGB as input following the original PIFuHD setting.
>
> It is non-trivial to adapt Oflow to an RGBD setting. Therefore, we use only RGB as its input.
>
> To provide a fair comparison, we conduct another experiment on the real world dataset 3DPW, which contains only RGB. Please see the general response part for results on real world dataset.
>
> ---
>
> ### [Limitations 1] The nature of FREDO’s supervision required
>
> See response to **[Weakness 2] Supervision of GT mesh and flow during training is too strong** and **[weakness 2] Supervision of GT mask and depth is too strong during inference**.
>
> ---
>
> ### [Limitations 1] Inputs to FREDO vs inputs to baselines
>
> See response to **[weakness 3] Clarification about baselines**.
>
> ---
>
> ### [Limitations 1] Novelty of FREDO
>
> See response to **[Weakness 1] Novelty of FREDO**.

---

> ### Comment · Reviewer_nToe · 2021-09-01
> **Maintaining original rating after rebuttal**
>
> I would like to thank the authors for their response and all the other reviewers for their detailed reviews as well. As indicated by other reviewers and given the nature of training data required for this method, I find the comparison against classical dynamic reconstruction works (e.g. DynamicFusion) very important. Moreover I don't believe the quality of the results themselves (especially considering the expensive training data required) or the added technical novelty on top of OFlow + PIFuHD merits acceptance of this paper to NeurIPS. Hence I would like to maintain my original rating.

---

> > ### Author Response · Authors · 2021-09-01
> > **Response to Reviewer nToe's Reply**
> >
> > We encourage the reviewer to reconsider the score based on the following points which we think clearly show the benefits of the proposed approach compared to prior work.
> >
> > ### Comparison to classical dynamic reconstruction works
> >
> > As stated in “**Compare to DynamicFusion**” in the general response, we cannot compare FREDO to it since there is no official code available.
> >
> > Instead, we compare FREDO to SurfelWarp [i], which outperforms DynamicFusion (see Fig. 7 and Fig. 8 in [i]). Even though the official implementation is available (https://github.com/weigao95/surfelwarp), we found there still exist unsolved bugs. After one week of debugging to fix code issues (e.g., out of bounds memory accesses) and to ensure that the comparison is fair (e.g., provide exactly identical input to SurfelWarp and our method), we are finally able to run SurfelWarp on SAIL-VOS 3D and we are confident that the obtained results are correct. Qualitative results can be found at: https://anonymous.4open.science/r/fredo_vis-D712/vis2.pdf (the webpage may take a while to load). We first elaborate on qualitative results (1) before providing more information about how we obtained the results (2).
> >
> > 1. **Qualitative results**: we illustrate meshes generated from point clouds produced by SurfelWarp. It is apparent that SurfelWarp lacks the ability to hallucinate unseen parts of objects, which is expected. Moreover, SurfelWarp cannot estimate motion correctly on SAIL-VOS 3D. For example, duplicated hands/feet are observed. This is because SAIL-VOS 3D contains relatively larger motion between images, making inference more challenging. Specifically:
> >
> >     - VolumeDeform scenes, which SurfelWarp is developed upon, contain 1200 frames (UPPER BODY), 552 frames (HOODIE), 876 frames (SUNFLOWER), and 727 frames (UMBRELLA) and only span several seconds (see Supplementary of VolumeDeform https://www.lgdv.tf.fau.de/files/2018/05/innmann2016deform_supplemental.pdf).
> >
> >     - SAIL-VOS 3D’s scenes are recorded at 9 frames per second.
> >
> > We think this clearly shows the benefits of the developed method and its ability compared to existing work.
> >
> > 2. **Implementation Details**:
> > - Note, there are unsolved bugs in the official code repo of SurfelWarp [i], as reported in the official issues listed below. We spent considerable time and effort to pinpoint and fix these problems:
> >     - https://github.com/weigao95/surfelwarp/issues/16
> >     - https://github.com/weigao95/surfelwarp/issues/34
> >     - https://github.com/weigao95/surfelwarp/issues/51
> >
> > - We also identified additional unreported issues (e.g., out of bounds memory accesses) which we fixed.
> >
> > - The VolumeDeform dataset that SurfelWarp [i] is developed upon is currently not available for download. See https://www.lgdv.tf.fau.de/publicationen/volumedeform-real-time-volumetric-non-rigid-reconstruction/. We have contacted the dataset authors and they confirmed in an email conversation that the server is down and that they do not have a copy anywhere else. To quote the email from the dataset authors:
> > > Sorry that the link does not work. I checked with the administrator of the site and unfortunately there are some technical problems … I don't have the data on a different location, so unfortunately I don't have any solution
> > >
> > Unfortunately, we cannot test on the VolumeDeform dataset at this point.
> >
> > - To ensure that the comparison is fair and rigorous, we adjusted SurfelWarp [i] to use GT masks and GT camera poses from SAIL-VOS 3D. Hence both our method and SurfelWarp [i] use exactly the same data. The comparison is hence fair.
> >
> > [i] Wei Gao and Russ Tedrake, "SurfelWarp: Efficient Non-Volumetric Dynamic Reconstruction." RSS. 2018.
> >
> > ---
> >
> >
> > ### Clarification of the quality of FREDO’s results
> >
> > We provide additional qualitative results on the real-world “3D person in the wild” (3DPW) dataset via this anonymous link: https://anonymous.4open.science/r/fredo_vis-D712/vis1.pdf (the webpage may take a while to load). We find the proposed method to outperform the other two baselines clearly. The proposed method is able to predict detailed, complete, and clean meshes of the dynamic objects (moving humans in this dataset).
> >
> > ---
> >
> >
> > ### Clarification of expensive training data
> >
> > We respectfully disagree. Because SAIL-VOS 3D is a synthetic dataset, the GT masks, meshes, and correspondences are not expensive to acquire. As stated in our earlier reply “**\[Weakness 2\] Supervision of GT mesh and flow during training is too strong**”, large-scale synthetic datasets have shown benefits for vision tasks.
> >
> > In addition to SAIL-VOS 3D we are aware of at least two academic projects ([j] and one more not yet published work) which are studying how to generate diverse, plausible synthetic data for dynamic objects. In addition, companies are moving in a similar direction, e.g.,
> >
> > - Unity’s platform for computer vision (https://unity.com/products/computer-vision) that provides “Diverse, affordable and unbiased synthetic data, perfectly labeled to train smarter computer vision models”;
> >
> > - NVIDIA’s Isaac Sim platform (https://developer.nvidia.com/isaac-sim) which is built on Nvidia Omniverse (https://www.nvidia.com/en-us/omniverse) that provides “physically accurate simulation” of “assets”.
> >
> > Even earlier, in addition to the SUNCG [a], ShapeNet [b], Playing for Benchmark [c] in our previous response, there are more synthetic datasets that have demonstrated effectiveness for various tasks, e.g., ModelNet [k], PartNet [l], SceneNet RGB-D [m].
> >
> > [j] Li et al., 4DComplete: Non-Rigid Motion Estimation Beyond the Observable Surface, ArXiv 2105.01905.
> >
> > [k] Wu et al., 3D ShapeNets: A Deep Representation for Volumetric Shapes. CVPR 2015.
> >
> > [l] Mo et al., PartNet: A Large-scale Benchmark for Fine-grained and Hierarchical Part-level 3D Object Understanding. CVPR 2019.
> >
> > [m] McCormac et al., SceneNet RGB-D: Can 5M Synthetic Images Beat Generic ImageNet Pre-training on Indoor Segmentation? ICCV 2017.
> >
> > ---
> >
> >
> > ### Clarification about the technical novelty
> >
> > As stated in previous response “**\[Weakness 1\] Straightforward combination of Oflow and PIFuHD**”, differs in key aspects from Oflow and PIFuHD: a canonical space for dealing with occlusion (Sec. 3.1), utilization of image features to provide more information about temporal coherence (L191-192), and a transformer to aggregate information from sequences (L164).  We think the following results can convincingly support the novelty of FREDO:
> >
> > - quantitative results of comparison to Oflow and PIFuHD on SAIL-VOS 3D in Tab. 1 (paper)
> > - detailed ablation study of FREDO in Tab. 2 (paper)
> > - quantitative results comparing to Oflow and PIFuHD on the real world dataset 3DPW in the general response of our feedback
> > - qualitative results comparing to Oflow and PIFuHD on the real world dataset 3DPW in the general response (https://anonymous.4open.science/r/fredo_vis-D712/vis1.pdf).

---

### Official Review · Reviewer_SY53 · 2021-07-16

**Rating:** 4
**Confidence:** 4

**Summary:**

This paper presents a supervised algorithm for 4D reconstruction of objects. The input to the system are ground truth instance masks, ground truth depth maps and RGB images. The system outputs a 3D model in a canonical space. The algorithm has three main components, first a feature extractor that convert 2.5D observations into dense feature vectors. Then, A flow field is learnt jointly, which maps observations in each view into the canonical space. Finally, a aggregation network is used to combine information from all observations, and generate an implicit field in canonical space.

The training signals used by the method are ground truth shapes and dense correspondences in 3D.

**Limitations And Societal Impact:**

This paper does not discuss about limitations of the method.

**Main Review:**

----

Originality(4/10)

This paper presents a new method to the challenging problem of 4D reconstruction, which is a new framework inspired by OccFlow and PiFU. The new components are the aggregation network, the flow network and the strategy of moving window for generating and advancing the canonical space.

 My major concern to the presented method, is that it does not discuss any non-deep learning method at all. For example, DynamicFusion [1] uses a commodity RGBD sensor that reconstructs dynamic scenes in real time. Of course, it has limitations: it could only handle moderate motion without topology changes (which are not addressed in this paper either), but it is still powerful enough to be the backbone of many real world applications. Therefore, missing such discussions and comparisons greatly hurts the originality of the presented method, since those methods are, essentially, class-agnostic 4D reconstructions from Monocular RGBD videos.  The authors claim that in L3 ,Compared to prior works, our problem setting is more realistic yet more challenging for two reasons: 1) partial observation 2) class-independent reconstruction. Those claims are not true since[1], compared to this work, tackles both challenges well and is more realistic than the presented work since it works on real world data with a RGBD sensor.

[1]Newcombe, Richard A., Dieter Fox, and Steven M. Seitz. "Dynamicfusion: Reconstruction and tracking of non-rigid scenes in real-time." CVPR 2015.

----

Quality(4/10)

I'm not convinced by the results of this paper. On the contrary, I believe the results presented in this paper indicates the presented method is missing important parts, prohibiting it from solving the actual problem.

First of all, this method takes **ground truth** depth and **ground truth** instance masks as input. Since the observed data is noise free, it is expected that the final reconstruction should at least have good quality when rendered at the input view ,which is not the case as shown in Fig.5, for the motorbike, the truck, the airplane, the person on the floor and the helicopter result. Why this is the case?  This also defeats the purpose of having bold numbers in the table, since if the method is having hard time reproducing the input GT observations, then it is essentially broken somehow. Also it shows that it is less likely to work when only estimated depth/ instance masks are used. This point is further demonstrated in L347, where the authors used GT flow-field. GT flow-field should, in theory produce near perfect results: isn't putting the gt depth into 3D space and deform them using the gt flow-field already gives you the result? What's missing here?

Secondly, since the input are GT signals, the problem reduces to 3D correspondence estimation. Registering deforming surfaces is not a new problem, and many solutions exist. For example, [1] uses a deformation grid for regularizing motion, together with ICP for registering global translations and rotations. How does this strategy compare to flow estimation presented in the paper?  Is Neural ODE style training of dynamics really necessary?

Thirdly, I wonder if the aggregation part really needs neural networks. Usually fusion can be done in many forms, the mostly used case is deforming a canonical space to the current view and fuse the observation using signed-distance functions. I understand the design choice in this paper is tailored to a MLP-based continuous occupancy field representation, but how much do you lose, or even gain, if switched to a discrete space with simple fusion strategy?

Finally, a minor issue is about the flow-field prediction. As I understand it, the flow-field is defined in the continuous space. The figures showed the in prediction (Fig(4), bottom right) seems to use the GT mesh for sampling the flow-field.  Shouldn't it be using the predicted surface instead?




---

Clarity(5/10)

There are some confusing descriptions which makes it hard to understand. For example:

1. L97:  instance masks and camera matrices -> does camera matrices mean intrinsic matrices? or the entire projection matrices?
2. the use of $\theta$ is a bit misleading: it is used both in v and g, but nowhere else. what does that mean?
3. Eq.2 is a bit misleading. According to Sec.3.2, there's nontrivial dependencies between each component: $\Phi$ depends on a integration of $\Phi$ and $v$, where $v$ depends on aggregated features. Though it makes the presentation simple to look at, it hides all the necessary details to understand the structure of the method.



---
Significance (3/10)

I'm negative about the significance of this paper. Major concerns are:

1. The setup is too simple. Assuming GT depth maps and instance masks is okay, but the quality of the results have to be much better to justify the setup

2. The quality is poor even the setup is way too realistic. Even with GT depth, mask and flow-field the method doesn't solve the problem near perfectly, which makes me believe there's something missing in the method or it is partially broken.

3. I don't see how this could generalize to more realistic setups, such as using a RGBD sensor as input, with estimated instance masks. RGBD sensors are noisy and instance segmentation are not perfect; I would expect the method to degrade even further to such setups, since it is supervised by synthetic data with Gt inputs.

4. Existing method already show real world examples of 4D reconstruction with a RGBD sensor, in real time. I'm not sure how this paper compare to that, or what it adds to the current progress of that problem.

**Time Spent Reviewing:**

5

---

> ### Author Response · Authors · 2021-08-10
> **Response to Reviewer SY53**
>
> ### [Originality] Compare to DynamicFusion
>
> Please refer to the general response for discussion about comparison to DynamicFusion.
>
> ---
>
> ### [Quality 1] Clarification about FREDO’s reconstruction quality at Input view
>
> Thanks a lot for bringing this up. We agree that given GT depth and masks, we can easily reconstruct a mesh that matches the depth map perfectly when projected to the same view.
>
> We did not enforce the implicit function to incorporate such a constraint. Therefore, the implicit function isn’t guaranteed to produce a geometry matching the input depth. To improve this we 1) use the input depth map to produce a geometry which matches the depth map when projected into the same view; we 2) use FREDO to only reconstruct geometry for $z$ values that aren’t visible in any view. This improves the performance of FREDO to 35.1 (+2.6) IoU and 1.09 (-0.18) Chamfer distance.
>
> ---
>
> ### [Quality 1] Clarification about Oracle experiment (L347)
>
> Sorry for the confusion. The intention of this oracle experiment is to study the accuracy of the predicted flow-field. During inference, to reconstruct N consecutive frames, we use the GT mesh of the center/canonical frame and propagate this mesh to other frames using the predicted flow-field. If GT flow-fields were used, we would achieve perfect reconstruction results for all N frames. In comparison, we obtained 72.6 IoU and 0.59 Chamfer distance indicating that the current flow-field estimate is good but could be further improved.
>
> ---
>
> ### [Quality 2] Comparison between deformation grid and our flow estimation
>
> Most importantly, we estimate scene flow at any continuous point in time $t$ irrespective of whether an image observation at this time is available. This is due to the fact that we only use an **aggregated** feature $\mathbf{z}_\mathbf{p}$ to compute the velocity field (Eq. (4)). Note, $\mathbf{z}_\mathbf{p}$ does not depend on any specific time step or visual frame. Instead, we use a Neural ODE [6] to integrate the velocity field to any specified time (Eq. (5)). In contrast, deformation grid estimate scene flow at **discrete** frames only because the corresponding image is needed.
>
> To validate the effectiveness of the proposed scene-flow, we provide the result of a scene-flow baseline in the main paper Sec. 4.4 Ablation 3 (L329-334, Tab. 2 “implicit flow”). The scene-flow of the baseline is directly estimated rather than being obtained through a velocity-field network. This baseline is similar to the scene-flow module used in [22].
>
> ---
>
> ### [Quality 3] Clarification about Fig. 4
>
> As discussed in response to **[Quality 1] Clarification about Oracle experiment (L347)**, we only study the differences between GT and predicted flow. To do so, we use the GT mesh to sample **predicted** flow field.
>
> ---
>
> ### [Clarity 1] Clarification about “camera matrices” in L97
>
> Camera matrices refer to entire projection matrices.
>
> ---
>
> ### [Clarity 2] Clarification about $\theta$
>
> As stated in L227, we use $\theta$ to subsume all trainable parameters. Specifically, $\theta$ contains trainable parameters for 1) temporal aggregator (Sec. 3.2.a); 2) feature extractor (Sec. 3.2.b); 3) velocity network $v$ (Sec. 3.2.c); and 4) reconstruction of canonical 4D implicit function $g$ (Eq. (1)).
>
> ---
>
> ### [Clarity 3] Clarification about Eq. 2
>
> With “$v$ depends on aggregated features”, we want to emphasize that aggregated features, i.e., $\mathcal{x}_\mathcal{p}$ in Eq. (2) and $\mathcal{z}_\mathcal{p}$ in Eq. (3), are not the same. Specifically, the input aggregated feature $\mathcal{z}_\mathcal{p}$ which is used to estimate the velocity field $v$ does not depend on the flow field $\Phi$ as stated in Eq. (4).
>
> ---
>
> ### [Significance 1 and 2] Clarification about setup is too simple and quality is not high
>
> See response to **[Quality 1] Clarification about FREDO’s reconstruction quality at Input view**.
>
> ---
>
> ### [Significance 3] Generalization to more realistic setups
>
> To show the robustness to estimated depth and masks of FREDO, we conduct experiments on real world data 3PWD. Please refer to the general response part for results on 3PWD.
>
> ---
>
> ### [Significance 4] Compare to existing methods
>
> See general response for comparing to DynamicFusion.

---

> ### Comment · Reviewer_SY53 · 2021-09-02
> **Maintain my current rating**
>
> I would like to thank the authors for their response.
>
> Dynamic Fusion sadly is lacking a public implementation, which makes it hard to compare against. A probably more convincing way to compare against Dynamic Fusion is to run the proposed method on some similar RGBD input sequences, for example, a slowly moving person recorded by an RGBD camera. It doesn't have to be an extensive quantitative experiment, it only has to be a small qualitative experiment, showing that the proposed method can work at least on par with dynamic fusion, in similar setups.
>
> Aside from the experiment, Dynamic Fusion surely deserves some in-depth discussion in the main text. The discussion in the main response is focused on differences, not the merits of the proposed method. For example, isn't a streaming-based optimization what we want in real life? Does optical flow give the proposed method an edge over dynamic fusion? (It should, yet I'm not convinced by the results on synthetic datasets shown in the paper.)
>
> Regarding the input view reconstruction, I'm proposing it as a sanity check of the method, not as an optimization goal using extra losses. The reasoning is just simple, if, given noise-free depth maps and camera poses, you should not be expecting much degradation when reconstructing the input depth. Imposing this as an extra goal is not a good solution, since the depth map will always be noisy in real life, this extra optimization goal might hurt instead of help if using real depth sensors.
>
> In general, I believe the task itself is important, and the authors are making efforts to solve it. But the presented solution seems half-baked and lacks convincing evidence.  I don't think the current solution makes significant progress towards solving the problem, therefore would vote for rejection.

---

> > ### Author Response · Authors · 2021-09-03
> > **Response to Reviewer SY53's Reply**
> >
> > Thanks a lot for agreeing that we address an important problem. We think our solution is elegant and we provide convincing evidence regarding its benefits. To make this concrete:
> >
> > ### Compare to prior dynamic reconstruction method
> >
> > We appreciate the reviewer’s understanding of the difficulty for comparing to DynamicFusion.
> >
> > Instead, we compare FREDO to SurfelWarp [a], which outperforms DynamicFusion (see Fig. 7 and Fig. 8 in [a]). Even though the official implementation is available (https://github.com/weigao95/surfelwarp), we found there still exist unsolved bugs. We fixed code issues (e.g., out of bounds memory accesses) and ensured that the comparison is fair (e.g., provide exactly identical input to SurfelWarp and our method). We are confident that the obtained results are correct. Qualitative results can be found at: https://anonymous.4open.science/r/fredo_vis-D712/vis2.pdf (the webpage may take a while to load). We first elaborate on qualitative results (1) before providing more information about how we obtained the results (2).
> >
> > 1. **Qualitative results**: we illustrate meshes generated from point clouds produced by SurfelWarp. It is apparent that SurfelWarp lacks the ability to hallucinate unseen parts of objects, which is expected. Moreover, SurfelWarp cannot estimate motion correctly on SAIL-VOS 3D. For example, duplicated hands/feet are observed. This is because SAIL-VOS 3D contains relatively larger motion between images, making inference more challenging. Specifically:
> >
> >     - VolumeDeform scenes, which SurfelWarp is developed upon, contain 1200 frames (UPPER BODY), 552 frames (HOODIE), 876 frames (SUNFLOWER), and 727 frames (UMBRELLA) and only span several seconds (see Supplementary of VolumeDeform https://www.lgdv.tf.fau.de/files/2018/05/innmann2016deform_supplemental.pdf).
> >
> >     - SAIL-VOS 3D’s scenes are recorded at 9 frames per second.
> >
> > We think this clearly shows the benefits of the developed method and its ability compared to existing work.
> >
> > 2. **Implementation Details**:
> > - Note, there are unsolved bugs in the official code repo of SurfelWarp [a], as reported in the official issues listed below. We spent considerable time and effort to pinpoint and fix these problems:
> >     - https://github.com/weigao95/surfelwarp/issues/16
> >     - https://github.com/weigao95/surfelwarp/issues/34
> >     - https://github.com/weigao95/surfelwarp/issues/51
> >
> > - We also identified additional unreported issues (e.g., out of bounds memory accesses) which we fixed.
> >
> > - The VolumeDeform dataset that SurfelWarp [a] is developed upon is currently not available for download. See https://www.lgdv.tf.fau.de/publicationen/volumedeform-real-time-volumetric-non-rigid-reconstruction/. We have contacted the dataset authors and they confirmed in an email conversation that the server is down and that they do not have a copy anywhere else. To quote the email from the dataset authors:
> > > Sorry that the link does not work. I checked with the administrator of the site and unfortunately there are some technical problems … I don't have the data on a different location, so unfortunately I don't have any solution
> > >
> > Unfortunately, we cannot test on the VolumeDeform dataset at this point.
> >
> > - To ensure that the comparison is fair and rigorous, we adjusted SurfelWarp [a] to use GT masks and GT camera poses from SAIL-VOS 3D. Hence both our method and SurfelWarp [a] use exactly the same data. The comparison is hence fair.
> >
> > [a] Wei Gao and Russ Tedrake, "SurfelWarp: Efficient Non-Volumetric Dynamic Reconstruction." RSS. 2018.
> >
> > ---
> >
> >
> > ### Clarification about comparing to DynamicFusion on slow moving person
> >
> > As stated in “**Compare to prior dynamic reconstruction method**”, we tried to compare to SurfelWarp on the VolumeDeform dataset. Unfortunately, it isn’t accessible at this time. Moreover, note that use of less dense data increases applicability of the developed method beyond settings where temporally dense data is available.
> >
> > For example, in **uncontrollable** video records,  “motion blur” is common for objects with relatively large motion recorded with consumer-level mobile cameras, whose video frame rates are usually 30 FPS. This “motion blur” will harm the reconstruction as high-quality correspondences across frames are inaccurate. Upon removing frames with apparent “motion blur” before reconstruction yields temporally sparse data even though the original video may be recorded at 30 FPS.
> >
> > We think this is important.
> >
> > ---
> >
> > ### Clarification about FREDO’s merits compared to DynamicFusion
> >
> > 1. As a learning-based method, FREDO can hallucinate the unseen parts of objects;
> > 2. FREDO can handle relatively large-motion compared to DynamicFusion-style works. These benefits are clearly apparent in https://anonymous.4open.science/r/fredo_vis-D712/vis2.pdf.
> >
> > Note, we do not use optical flow in FREDO. We assume the reviewer refers to the flow field we mentioned in the paper. We want to emphasize that we have shown that the flow field contributes to FREDO’s performance (see L329 - 337).
> >
> > ---
> >
> > ### Clarification about streaming-based vs batch-based
> >
> > We respectfully disagree that “streaming-based optimization is what we want in real life”. Instead, we think streaming-based and batch-based methods serve different purposes. For many tasks a batch-based setting is more suitable. For example, 1) game asset creation for articulate objects, which is extremely expensive with current technology; 2) adding entertaining 3D visual effects to consumer video clips, e.g., for 3D visual effects in TikToks, batch-based methods are more suitable.
> >
> > ---
> >
> >
> > ### Clarification about input view reconstruction
> >
> > Please note that in “**Robustness and generalizability: test on real world dataset**” of the general response, we do not use the input view sanity check.
> >
> > Quantitatively, FREDO outperforms both baselines PIFuHD and Oflow by a big margin in this setting.
> >
> > Qualitatively, we find the proposed method to outperform the other two baselines clearly. The proposed method is able to predict detailed, complete, and clean meshes of the dynamic objects (moving humans in this dataset). Please see https://anonymous.4open.science/r/fredo_vis-D712/vis1.pdf (the webpage may take a while to load).

---

### Official Review · Reviewer_Jqx8 · 2021-07-17

**Rating:** 3
**Confidence:** 4

**Summary:**

The paper proposed a neural network architecture for reconstructing shape and motion from RGBD sequence. It mainly consists of two modules, (i) infer a canonical space which permits to aggregate visual cues across time. (ii) a 4D transformation module to capture correspondences.  The experiment is done on a synthetic dataset.

**Main Review:**

The paper tries to solve for a challenging problem, i.e. reconstruct shape and motion from RGBD sequence, which in literature we tend to refer as “dynamic fusion”. The best paper in CVPR15 by Newcombe et al. is perhaps one of the most famous works which try to solve this task. But the authors somehow neglect this line of works, which raises questions of the credibility of the paper. Moreover, the title of the paper is misleading, “... from Monocular videos” by convention refers to using video captured by a single rgb camera, which is an even more challenging setup compared to what is actually used in this paper, i.e. rgbd sensor. In addition, the result presented in this paper looks far less desirable, especially when compared to the visual result presented in the dynamic fusion paper. Also the experiment is only conducted on synthetic data, and the result for non-human subject does not look good enough to back up the “class agnostic” claim.

My other critics / comments are:
1) Page 3 Ln. 77 when discussing NRSfM, missing reference to a few most classical paper, e.g. “A Simple Prior-free Method for Non-Rigid Structure-from-Motion Factorization”, Dai te al. CVPR12.

2) Ln. 79, it claims “does not rely on any templates or priors”. I think this is a overclaim, as this work learns from a synthetic dataset, thus actually uses a data-driven prior, which is no less when compared to what is used in NRSfM.

3) I am not fully convinced by the proposed framework. The convention wisdom is that solving the 3D scene flow and 3D shape is a joint task, as both modules should help each other. However, it seems that in this paper, the flow field inference module is sort of an independent process. I think the authors should try to polish their method, and improve the result to be at least comparable to the real state-of-the-arts (e.g. Newcombe et al.).


**Time Spent Reviewing:**

4

---

> ### Author Response · Authors · 2021-08-10
> **Response to Reviewer Jqx8**
>
> ### [Main Review] Not cite DynamicFusion
>
> We have tried our best to reproduce DynamicFusion’s results before the submission and can neither find suitable code nor were we able to fix existing code (please refer to the general response for discussion about DynamicFusion). Meanwhile, note that DynamicFusion cannot provide the same level of performance as learning-based works, e.g., [a]. Unfortunately, NDG [a] also didn’t provide their own implementation of DynamicFusion. For above reasons, we did not compare to DynamicFusion in the submission and we follow recent learning-based papers. We will add a discussion in the revised version.
>
> [a] Bozic et al., Neural Deformation Graphs for Globally-consistent Non-rigid Reconstruction, CVPR 2021.
>
> ---
>
> ### [Main Review] Clarification of “monocular”
>
> Sorry for the confusion. We will make the title more concrete in the revised version.
>
> ---
>
>
> ### [Main Review] Experiment is only conducted on synthetic dataset
>
> To demonstrate the effectiveness of FREDO, we conduct an experiment on a real world dataset 3DPW. Please refer to the general part for results on 3DPW.
>
> ---
>
>
> ### [Comments 1] Missing reference
>
> Thanks a lot. We will add.
>
> ---
>
>
> ### [Comments 2] Clarification of “not rely on any template or priors”
>
> For “template or prior”, we mainly refer to 2D key points used in [3, 31, 51], which are hard to acquire and require additional human annotation. In contrast, FREDO does not need such information.
>
> ---
>
>
> ### [Comments 3] Clarification of whether FREDO is jointly-trained
>
> We agree that it is beneficial to solve 3D scene flow and 3D shape together. As stated in L228, FREDO is trained to jointly optimize the shape and temporal losses (Eq. (6)) concurrently.

---

> > ### Comment · Reviewer_Jqx8 · 2021-08-23
> > **Response to the authors**
> >
> > I respectfully disagree with the author's response regarding to comparisons, and lean to maintain my current rating.
> >
> > The "DynamicFusion" work is just one example of RGBD fusion approaches which the paper should compare. There are in fact several followup works whose official implementation are available online, e.g. Killingfusion, SurfelWarp (https://github.com/weigao95/surfelwarp).
> > The authors are of course correct in that their approach is different, but there's no evidence in the paper showing those differences actually leads to improvement.
> >
> > The authors also argue that learning method already surpassed dynamic fusion methods by referring [a], which I respectfully disagree. Pls note that this paper is proposing a "class agonostic" approach, thus citing [a] which uses  human SMPL model is not relevant to the argument. Moreover, the visual quality of the proposed approach on human subjects are noticablly less desirable compared to [a] and other recent human reconstruction works.
> >
> > The author states that the method is jointly trained with shape and temporal loss, but this does not answer  my critic that the method may be inefficient in utilizing shape and temporal information at inference time.
> >
> > Finally, the intro and releted works needs major rewrite, as the corrent version has overclaim issues, and misses discussion to dynamic RGBD fusion works.

---

> > > ### Author Response · Authors · 2021-08-31
> > > **Response to Reviewer Jqx8's Reply**
> > >
> > > ### Clarification about Learning method already surpassed DynamicFusion
> > >
> > > NDG [a] shows that learning-based methods improve upon DynamicFusion. **Importantly, different from the reviewer’s claim, NDG [a] doesn’t use the human SMPL model.** We never stated anything different. Indeed, NDG requires the input to be a signed distance field (SDF) grid. To obtain the grid, NDG requires “four calibrated cameras” and converts four depth maps to an SDF with “static volumetric reconstruction” (see Sec. 3.1 of the NDG paper). In contrast, FREDO assumes a single-camera RGB-D input per frame which prevents a direct comparison.
> > >
> > > Different from NDG [a], we focus on the single-camera RGB-D setting which is arguably more challenging than the “four calibrated camera” setting of NDG [a]. We think a learning-based method is even more suitable in addressing these additional challenges.
> > >
> > > ---
> > >
> > > ### Compare to prior dynamic reconstruction approaches
> > >
> > > **Regarding KillingFusion**: there isn't an official implementation. The most-starred repo (https://github.com/saurabheights/KillingFusion) lacks several critical components, such as “Rigid Registration” as the author specifies in the README (see “What needs work” in https://github.com/saurabheights/KillingFusion/blob/1076239/README.md).
> > >
> > > **Regarding SurfelWarp**: After one week of debugging to fix code issues (e.g., out of bounds memory accesses) and to ensure that the comparison is fair (e.g., provide exactly identical input to SurfelWarp and our method), we are finally able to run SurfelWarp on SAIL-VOS 3D. Qualitative results can be found at: https://anonymous.4open.science/r/fredo_vis-D712/vis2.pdf (the webpage may take a while to load). We first elaborate on qualitative results (1) before providing more information about how we obtained the results (2).
> > >
> > > 1. **Qualitative results**: we display meshes generated from point clouds produced by SurfelWarp. It is apparent that SurfelWarp lacks the ability to hallucinate unseen parts of objects. Moreover, SurfelWarp cannot estimate motion correctly on SAIL-VOS 3D. For example, duplicated hands/feet are observed. This is because SAIL-VOS 3D contains relatively larger motion between images, making inference more challenging. Specifically:
> > >
> > >     - VolumeDeform’s scenes, which SurfelWarp is developed upon, contain 1200 frames (UPPER BODY), 552 frames (HOODIE), 876 frames (SUNFLOWER), and 727 frames (UMBRELLA) and only span several seconds (see Supplementary of VolumeDeform https://www.lgdv.tf.fau.de/files/2018/05/innmann2016deform_supplemental.pdf).
> > >
> > >     - SAIL-VOS 3D’s scenes are recorded at 9 frames per second.
> > >
> > > We think this clearly shows the benefits of the developed method and its ability compared to existing work.
> > >
> > > 2. **Implementation Details**:
> > > - Note, there are unsolved bugs in the official code repo of SurfelWarp, as reported in the official issues listed below. We spent considerable time and effort to pinpoint and fix these problems:
> > >     - https://github.com/weigao95/surfelwarp/issues/16
> > >     - https://github.com/weigao95/surfelwarp/issues/34
> > >     - https://github.com/weigao95/surfelwarp/issues/51
> > >
> > > - We also identified additional unreported issues (e.g., out of bounds memory accesses) which we fixed.
> > >
> > > - The VolumeDeform dataset that SurfelWarp is developed upon is currently not available for download. See https://www.lgdv.tf.fau.de/publicationen/volumedeform-real-time-volumetric-non-rigid-reconstruction/. We have contacted the dataset authors and they confirmed in an email conversation that the server is down and that they do not have a copy anywhere else. To quote the email from the dataset authors:
> > > > Sorry that the link does not work. I checked with the administrator of the site and unfortunately there are some technical problems … I don't have the data on a different location, so unfortunately I don't have any solution
> > > >
> > > Unfortunately, we cannot test on the VolumeDeform dataset at this point.
> > >
> > > - To ensure that the comparison is fair and rigorous, we adjusted SurfelWarp to use GT masks and GT camera poses from SAIL-VOS 3D. Hence both our method and SurfelWarp use exactly the same data.
> > >
> > > ---
> > >
> > >
> > > ### Clarification about “utilizing shape and temporal information at inference time”
> > >
> > > We do not think an argument like “the method may be inefficient in utilizing shape and temporal information” is a valid reason to reject a paper. We find the current method to yield compelling results as we showed through numerous results and ablations. Similar to any other research, the proposed method will likely be improved upon in future work. This is to be expected.

---

### Official Review · Reviewer_7c6w · 2021-07-18

**Rating:** 4
**Confidence:** 4

**Summary:**

This paper recovers dense 3D shape of a possibly articulating and occluded object given a RGB-D video with known instance masks and camera intrinsics/extrinsics. The paper proposes an approach that relies on two learned modules:

a) An implicit velocity field v(p,t) which predicts velocity at point p at time t, using additional image based features for the location p across all timesteps.
b) A canonical space occupancy f(p), which aggregates pixel-aligned features from the various frames using projection of 3D coordinates obtained by integrating the velocity field (essentially a generalization of PixelNeRF and PIFu, but with a velocity-field).


The approach relies on strong supervision in the form of known 3D shapes as well as dense 3D scene flow (so both the above modules can be directly supervised) and presents results on 5 categories using data from GTA-V.


**Limitations And Societal Impact:**

I think more discussion on the limitations e.g. what categories can be handled, and the strong supervision required, might be beneficial.

**Main Review:**

On the positive side, I really like the formulation using the implicit velocity field to model motion. This is a simple and elegant idea which was also used in the OccupancyFlow work, but differs from some other recent approaches which directly predict canonical-to-view deformations at each time-step instead of predicting a velocity field. In context of this, this paper introduces some novel deisgn and architecture choices, and one that future efforts could build upon e.g. integrating pixel-aligned features, and predicting velocity using features from across timesteps - both sensible choices that can help generalization.

Further, the empirical results do clearly show improvements over static-scene prediction methods, thus demonstrating the benefits of using multiple video frames (although I am not sure if the baselines were modified to provide depth as additional input)
.
However, although aspects of the technical approach are novel, I would not argue for acceptance. In particular:

a) The approach requires very strong supervision for learning i.e. full 3D across timesteps as well as 3D scene flow are required, and even at inference, calibrated RGB-D videos with instance masks are needed. I think this requirement makes the approach not very scalable, and this is in contrast to recent work e.g. NeRFies, NSFF etc which accomplish similar goals, but without such strong training/test time annotations.

b) While the paper claims to perform "class-agnostic" reconstruction, the experiments are really on a limited set of categories and I suspect the model does infact rely on category-level priors. To support claims of category-agnostic reconstruction, the work should report experiments where the learned model can generalize to new categories.

c) On a note related to the above, I think the results shown are not of a very high-quality, specially for non-human categories (e.g. the helicopter in main text, truck in appendix)

d) Assuming calibrated RGB-D videos and instance masks is a rather strong supervision, and I would be curious to see comparisons against recent NeRF based methods (e.g. NeRFies can easily to use depth and flow supervision) as well as more classical methods e.g. DynamicFusion.

e) As alluded to above, the key contributions of this work are to integrate PIFu-like design choices into the OccupancyFlow framework. While this is certainly desirable, I do not think it is a significant contribution on its own.

**Time Spent Reviewing:**

4

---

> ### Author Response · Authors · 2021-08-10
> **Response to Reviewer 7c6w**
>
> ### [Main review] Supervision of the GT mesh and flow
>
> Three trends led to the decision to study the use of ground truth meshes and flows for the challenging task of non-rigid dynamic object-agnostic mesh reconstruction. 1) Supervised learning has been shown again and again to be remarkably successful. In fact, most 3D reconstruction works (e.g., ONet [26], PIFu [36], etc.) are trained using GT synthetic meshes and then generalized to real-world data. 2) Recent advances have shown the benefits of large-scale synthetic data, e.g., SunCG [a], ShapeNet [b], Playing for Benchmark [c], etc. 3) The ability of transfer learning has also been verified in recent synthetic platforms, e.g., embodied AI [d], optical flow [e], and autonomous driving [f] etc.
>
> Moreover, large-scale real world data for non-rigid dynamic object-agnostic mesh reconstruction isn’t available. Instead, people either apply costly motion capture systems to limited dynamic object categories [g] or utilize 2D supervision and only evaluate on 2D metrics [h]. Instead, we conduct experiments on large-scale 3D datasets and evaluate the results rigorously using perfect pixel-aligned 3D GT.
>
> [a] Song et al., Semantic Scene Completion from a Single Depth Image. CVPR 2017.
>
> [b] Chang et al., ShapeNet: An Information-Rich 3D Model Repository. CVPR 2015.
>
> [c] Richter et al., Playing for Benchmarks. ICCV 2017.
>
> [d] Savva et al., Habitat: A Platform for Embodied AI Research. ICCV 2019.
>
> [e] Butler et al., A Naturalistic Open Source Movie for Optical Flow Evaluation. ECCV 2012.
>
> [f] Dosovitskiy et al., CARLA: An Open Urban Driving Simulator. CoRL 2017.
>
> [g] Slavcheva et al., KillingFusion: Non-rigid 3D Reconstruction without Correspondences. CVPR 2017.
>
> [h] Yang et al., LASR: Learning Articulated Shape Reconstruction from a Monocular Video. CVPR 2021.
>
> ---
>
>
> ### [Main review] Strong assumptions on GT camera, mask, and depth
>
> We agree that the use of GT information as input is a strong assumption. However, we think use of ground truth is insightful to the community when developing an end-to-end trainable framework for recovering a pixel-aligned representation of non-rigid dynamic objects which may only be partially visible within a frame. By ignoring input noise and by splitting development into manageable pieces, we can reveal the importance of individual parts. For instance, 1) accurate camera data isolates the development of temporal propagation and fusion (Sec. 3.2a) and a pixel-aligned representation (Sec. 3.2b) from other factors; 2) GT depth and mask let us focus on developing a canonical 4D implicit function (Sec. 3.1). Through experiments, we validated the effectiveness of individual modules for the task of 4D reconstruction (Sec. 4.4).
>
> In practice, these assumptions can be relaxed for real-world data. To demonstrate the robustness of FREDO to non-GT inputs, we evaluate FREDO on a real world dataset which only contains RGB sequences. Please see the general responses for more details.
> Please refer to the general part for experiments on real world dataset, demonstrating robustness of FREDO .
>
> ---
>
> ### [Main review] Compare to NeRFies, NSFF
>
> We respectfully disagree that either NeRFies or NSFF has stronger scalability than FREDO. Essentially, both NeRFies and NSFF require scene-specific training. Namely, a separate model needs to be trained for each video. This prohibits large-scale usage. Therefore, it is not tractable to run NeRFies on SAIL-VOS 3D. Meanwhile, because NeRFies do not provide the data they used in their publication, we cannot run FREDO on their data either.
>
> ---
>
> ### [Main review] Compare to DynamicFusion
>
> Please refer to the general part for discussion about comparing to DynamicFusion.
>
> ---
>
> ### [Main review] Contributions of FREDO
>
> FREDO has several key differences compared to Oflow: 1) we propose a canonical space representation to tackle object’s occlusion (Sec. 3.1) that Oflow does not take into account; 2) we utilize image features to provide more information about temporal coherence, e.g., discussion in L191-192 about $\mathbf{z}_\mathbf{p}$. Without such information, performance decreases from 32.5 to 26.4 (IoU) (L324-328); 3) we use a transformer to aggregate information from sequences (L164). This design improves IoU from 29.7 to 32.5 compared to vanilla average pooling (L316-323).
>
> Compared to PIFuHD, FREDO 1) develops a canonical space representation; 2) we consider dynamic objects instead of static ones.

---

> ### Comment · Reviewer_7c6w · 2021-08-19
> **Thanks for Response**
>
> I'd like to thank the authors for their response. In particular, I appreciate the fact the DynamicFusion is unfortunately not easy to reimplement and is not available off-the-shelf to compare. That said, I do not think most of the other concerns are well-addressed -
>
> Lack of Scalability due to Required Supervision:
>
> The response essentially offers 3-part argument: a) supervised learning works, b) large-scale synthetic data is growing, and c)transfer learning/domain adaptation can help. While I obviously agree with 'a', unfortunately (b,c) require a leap of faith when it comes to dynamic data - it is not obvious that diverse plausible synthetic data will be available for generic objects (simulating synthetic cats is hard!), or that transfer learning will work well.
> Unfortunately, I am not willing to make this leap of faith without empirical evidence - If the paper had show results on generic objects (not just humans) in real videos (through any training data), I would have been been more supportive.
>
> Nerfies, NSFF:
>
> I appreciate the point that these solutions may not be scalable, but one other question is whether they work better. Assuming one is ok with spending 12-GPU hours on a single sequence, would the proposed solution be better, or work like NSFF, Nerfies? Because these prior methods have shown results on real videos of generic objects, I am inclined to believe they might work better, and unless there is evidence otherwise.
>
> Quality of Results and Generalization:
>
> To reiterate,  the results shown are not of a very high-quality, specially for non-human categories. Further, to support claims of "category-agnostic" reconstruction, the work should report experiments where the learned model can generalize to new categories - I do not think the current system is capable of this.
>
> Contributions:
>
> As also alluded to by "Reviewer nToe", the paper's contribution is to combine insights from both OFlow and PIFu. By virtue of this combination, the work is obviously different from either of the two (which the response highlights), the critique is that this contribution is not significant enough on its own to argue for acceptance for acceptance.
>
> Overall, I would like to persist in the current rating. While I like the goal of the project, I do not think the current solution (strong reliance on test-time annotations, and even more stringent training time requirements)

---

> > ### Author Response · Authors · 2021-08-20
> > **Response to Reviewer 7c6w's Reply**
> >
> > ### Lack of Scalability due to Required Supervision
> >
> > We appreciate that the reviewer agrees with our point that supervised learning is effective. However, we kindly disagree that a leap of faith is required regarding the mentioned points (b) and (c):
> >
> > **Regarding large-scale synthetic data is growing**: We think the leap of faith is rather small when it comes to dynamic data. In addition to SAIL-VOS 3D we are aware of at least two academic projects ([i] and one more not yet published work) which are studying how to generate diverse, plausible synthetic data for dynamic objects. In addition, companies are moving in a similar direction, e.g.,
> > 1. Unity’s platform for computer vision (https://unity.com/products/computer-vision) that provides “Diverse, affordable and unbiased synthetic data, perfectly labeled to train smarter computer vision models”;
> >
> > 2. NVIDIA’s Isaac Sim platform (https://developer.nvidia.com/isaac-sim) which is built on Nvidia Omniverse (https://www.nvidia.com/en-us/omniverse) that provides “physically accurate simulation” of “assets”.
> >
> > Even earlier, in addition to the SUNCG [a], ShapeNet [b], Playing for Benchmark [c] in our previous response, there are more synthetic datasets that have demonstrated effectiveness for various tasks, e.g., ModelNet [j], PartNet [k], SceneNet RGB-D [l].
> >
> > **Regarding transfer learning**: In section **Robustness and generalizability: test on real world dataset** of the general response, we have shown FREDO successfully transfers from synthetic data (SAIL-VOS 3D) during training to real world data (3DPW) during testing. We think our result is aligned with [e.g., d, e, f]: a model trained on synthetic data can be transferred to real world scenarios.
> >
> > Specifically, on 3DPW, FREDO outperforms both static reconstruction (PIFuHD) and dynamic reconstruction (Oflow) baselines in all three metrics by a large margin, verifying the effectiveness of the proposed framework:
> >
> > 1. for IoU, FREDO yields 46.2 IoU while PIFuHD and Oflow achieve 42.3 and 39.7 respectively;
> >
> > 2. for Chamfer distance, FREDO’s 0.167 result outperforms both PIFuHD (0.242) and Oflow (0.265);
> >
> > 3. for L2-correlation error, FREDO provides 0.203, which is better than Oflow’s 0.374. Note, PIFuHD does not have the ability to model dynamics.
> >
> >
> > [i] Li et al., 4DComplete: Non-Rigid Motion Estimation Beyond the Observable Surface, ArXiv 2105.01905.
> >
> > [j] Wu et al., 3D ShapeNets: A Deep Representation for Volumetric Shapes. CVPR 2015.
> >
> > [k] Mo et al., PartNet: A Large-scale Benchmark for Fine-grained and Hierarchical Part-level 3D Object Understanding. CVPR 2019.
> >
> > [l] McCormac et al., SceneNet RGB-D: Can 5M Synthetic Images Beat Generic ImageNet Pre-training on Indoor Segmentation? ICCV 2017.
> >
> > ---
> >
> > ### Nerfies, NSFF
> >
> > We respectfully disagree.
> >
> > First, the task of free-view synthesis of dynamic scenes (NSFF/Nerfies) is different and simpler than 4D mesh reconstruction (ours). Essentially, we aim to reconstruct the **complete mesh** for generic objects rather than mostly a visible front surface. Moreover, our model is unified and doesn’t require scene-specific training. In fact, NSFF/Nerfies are not applicable to the dataset we used (SAIL-VOS 3D) which contains hundreds of different scenes.
> >
> > Second, we want to emphasize that Nerfies and NSFF are prohibitively costly to train and the training cannot be completed in “12-GPU hours” as claimed by the reviewer. Specifically, **for each scene**:
> >
> > 1. **Nerfies** requires $8 \text{(GPU cards)} \times 24 \text{(h)} \times 7 \text{(days)} = 1344$ GPU hours for the full model. See “We train on 8 V100 GPUs for a week for full HD models” (Sec. 5.1 in Nerfies).
> >
> > 2. **NSFF** requires $2 \text{(cards)} \times 24 \text{(h)} \times 2 \text{(days)} = 96$ GPU-hour training. See “Training a full model takes around two days per scene using two NVIDIA V100 GPUs” (Sec. 4 in NSFF).
> >
> > This prevents both methods from being used in large-scale settings and hence a reasonable comparison to FREDO. To make this concrete: in our setting, we use 2761 scenes from SAIL-VOS 3D. Using NSFF would require $2761 \text{(\\#scenes)} \times 96 \text{(NSFF training time)} = 265,056$ GPU hours (approx. 30 GPU years) to compute a single entry in our table. Even if the reviewer were to provide the funding and resources, the environmental impact is non-negligible.
> >
> > Moreover, **regarding inference time**, for FREDO, the total time to generate the mesh for one object is 1.07s (0.29s inference + 0.78s marching cube). In contrast, NSFF’s “rendering takes roughly 6 seconds for each 512 $\times$ 288 frame” (Sec. 4 in NSFF) and this time does not include the time to extract the mesh (e.g., marching cube) from the implicit model.
> >
> > ---
> >
> >
> > ### Quality of Results and Generalization
> >
> > **Regarding quality**, please see **[Quality 1] Clarification about FREDO’s reconstruction quality at Input view** in response to **reviewer SY53**. The vanilla FREDO doesn’t **enforce** the depth map to be faithfully reproduced. In an additional experiment, we 1) use the input depth map to produce a geometry which matches the depth map when projected into the same view and we 2) use FREDO to only reconstruct geometry for values that aren’t visible in any view. This improves the performance of FREDO to 35.1 (+2.6) IoU and 1.09 (-0.18) Chamfer distance.
> >
> >
> > **Regarding generalization**, as stated in L79, FREDO does not rely on any templates or priors similar to NR-SFM [3, 31, 51]. A unified FREDO model can be used to reconstruct different object categories, the results of which are provided in Tab. 1.
> >
> > ---
> >
> >
> >
> > ### Contributions of FREDO
> >
> > We appreciate that the reviewer agrees that FREDO is “obviously different from either of the two”.
> > However, we respectfully disagree that the contributions of FREDO are not significant.
> > We have clearly verified the importance and effectiveness of FREDO’s design choices in Tab. 2. Specifically,
> >
> > 1. without canonical space representation (Sec. 3.1), which Oflow does not consider, object occlusions can’t be tackled;
> >
> > 2. without utilizing image features (L324-328), we do not have enough information about temporal coherence and the performance decreases from 32.5 to 26.4 (IoU);
> >
> > 3. without a transformer for aggregating temporal information (L316-323), we cannot estimate the flow field from sequences reliably (L164) and the performance of IoU drops from 32.5 to 29.7.
> >
> > We want to emphasize that class-agnostic dynamic object reconstruction is a very difficult task and we think as a first step towards this goal, FREDO provides promising results.
> >
> > ---
> >
> >
> > ### Clarification about “strong reliance on test-time annotations”
> >
> > We respectfully disagree. We have demonstrated in the **Robustness and generalizability: test on real world dataset** section of the **general response** that FREDO does not require GT depth and mask during inference time and still outperforms existing baselines by a large margin.
> >
> > ---
> >
> > ### Clarification about “even more stringent training time requirements”
> >
> > For FREDO training time, please see Sec A.1 in the Appendix. We think training for 30 hours on 4 V100 is fairly affordable and faster than training of many other recent deep net architectures.

---

### Author Response · Authors · 2021-08-10
**General responses**

We thank all reviewers for detailed and constructive comments.

---

### Note on the references

In this material, we use [number] to refer to the related work in the submission, and [character] for new citations. We thank the reviewers for suggesting relevant work which we’ll incorporate in the revision.

---

### Compare to DynamicFusion

We tried our best to compare to DynamicFusion and we’ll continue to do so. Please take note of the following:

1. FREDO as a learning-based data-driven approach differs in key aspects from SLAM-based DynamicFusion:
    - **task setting**: DynamicFusion considers a stream-based reconstruction while FREDO focuses on a batch setting. Specifically, DynamicFusion observes input frames one by one while FREDO obtains all input frames.
    - **flow estimation**: DynamicFusion discritizes the non-rigid motion-fields into a predetermined set of deformation nodes (similar to voxels). In contrast, FREDO incorporates a continuous representation, i.e., implicit network, which predicts a velocity field for **arbitrary** points in canonical space (Eq. (4)). In principle, ours could be generalized to higher-resolution space and model finer motion details.
     - **input**: The input of DynamicFusion is solely the depth map of each object, while we use RGBD videos. The color information could help learn smooth articulation and consistent temporal transformation of objects.
     - **reconstruction**: as evidenced in qualitative results provided by DynamicFusion (https://youtu.be/i1eZekcc_lM), DynamicFusion cannot complete the backside of the object if the camera only moves in the front. In many cases (e.g., short video clips), the reconstructed meshes are merely the front-view surfaces even for highly symmetric geometric shapes. In contrast, as a learning-based model, FREDO is able to complete the geometry of the observed objects as shown in the multi-view visualization of Fig. 5 main paper and Fig.S2/S3 in our Appendix.

2. Due to patent issues or the authors choice, there is no official code available for DynamicFusion. Unfortunately, it turned out to be very hard to reproduce DynamicFusion results for the entire community. We have tried our best to find, fix and modify third-party-reimplementations. However, after diving into each repo, we find there does not exist a suitable one:
    - The most-starred repo (https://github.com/mihaibujanca/dynamicfusion) does not implement a warp field estimation let alone surface fusion (see report on https://github.com/mihaibujanca/dynamicfusion/blob/master/Report.md). Therefore, it cannot be used to reconstruct either static or dynamic objects.
    - Another version (https://github.com/swarth100/dynfu) does not implement surface fusion (see Sec. 6.7 of the project report in https://spina.me/assets/files/dynamic-fusion.pdf). This means they do not update geometry in the canonical space. Thus the repo is not able to reconstruct non-rigid objects.
    - The third-party-contributed OpenCV-contrib module (https://github.com/opencv/opencv_contrib/tree/master/modules/rgbd) does not have a correct implementation for the dense warp field interpolation (https://github.com/opencv/opencv_contrib/pull/2719). Therefore, the non-rigid object’s motion estimation is incorrect.
    - Neural Deformation Graphs (NDG) [d] compares to their self-implemented DynamicFusion. However, they have not released their implementation of DynamicFusion (https://github.com/AljazBozic/NeuralGraph/issues/1#issuecomment-866293129).

---

### Robustness and generalizability: test on real world dataset

To demonstrate the effectiveness of FREDO on real world data, and to alleviate the GT assumption of depth and instance masks, we conduct additional experiments. We use FREDO to reconstruct meshes on the 3DPW dataset [a]. 3DPW is a real-world dataset which contains RGB video sequences for people only. To obtain depth and instance masks, we utilize Consistent Video Depth Estimation (CVD) [b] for monocular depth estimation and Mask RCNN from Detectron2 [c] for instance segmentation. The dataset itself doesn’t provide GT mesh. Instead it labels human 3D key-points, which we feed into the SMPL template for generating GT meshes. Following the same experimental setting in our paper, we adopt PIFuHD as the state-of-the-art static reconstruction baseline and Oflow as the dynamic reconstruction baseline.

**Quantitative results**: we obtain the following results summarized in the table below. We observe the proposed method to significantly improve upon both baselines.

|Metrics|IoU|Chamfer|L2-corr|
| --------- | ------- | ------ | ------- |
| PIFuHD| 42.3  |0.242|  N/A  |
| Oflow   |39.7   |0.265|0.374 |
| ours     | 46.2  |0.167|0.203 |

**Qualitative results**: we provide additional qualitative results via this anonymous link: https://anonymous.4open.science/r/fredo_vis-D712/vis1.pdf (the webpage may take a while to load). We find the proposed method to outperform the other two baselines clearly. The proposed method is able to predict detailed, complete, and clean meshes of the dynamic objects (moving humans in this dataset).

[a] Marcard et al., Recovering Accurate 3D Human Pose in The Wild Using IMUs and a Moving Camera, ECCV 2018  (https://virtualhumans.mpi-inf.mpg.de/3DPW/)

[b] Luo et al., Consistent Video Depth Estimation, SIGGRAPH 2021 (https://github.com/facebookresearch/consistent_depth)

[c] Detectron2 (https://github.com/facebookresearch/detectron2)

---

### Decision · Program_Chairs · 2021-09-28

**Decision:**

Accept (Poster)

**Comment:**

This submission received four negative reviews from expert reviewers, who all chose to main their negative recommendations.  The shared concern is the lack of discussion and comparison with multiple related methods, in particular those along the line of DynamicFusion.  While the AC understands the authors' argument on the lack of publicly released code, the AC agrees with the reviewers that given the large number of follow-up methods and the similarities between the approaches, such comparisons are crucial, and the authors should have included a thorough comparison. The authors are encouraged to revise the paper based on the reviews.

**Consistency Experiment:**

NeurIPS has a long history of experimentation. In 2014, NeurIPS ran an experiment in which 10% of submissions were reviewed by two independent committees to quantify the randomness in the review process. This year, we repeated a variant of this experiment to see how the quality of the review process has changed over time.  This paper was part of the experiment and was therefore assigned to two committees (consisting of reviewers, an Area Chair, and a Senior Area Chair) that reached independent decisions.  If both committees made the same recommendation, this recommendation was followed. If a single committee recommended acceptance, the paper was accepted (with the exception of a few cases in which the other committee identified what we considered a fatal flaw, e.g., an error in a key result).

This copy’s committee reached the following decision: **Reject**

The other committee assigned to the paper recommended **Accept (Poster)**.  You can find the other set of reviews, along with any follow up discussion with the authors here:
https://openreview.net/forum?id=OP6ihHjllEc